# Control of clathrin-mediated endocytosis by NIMA family kinases

**Braveen B. Joseph**[1‡], **Yu Wang**[2,3‡], **Phil Edeen**[1], **Vladimir Lažetić**[1], **Barth D. Grant**[2], **David S. Fay**[1]*

**1** Department of Molecular Biology, College of Agriculture and Natural Resources, University of Wyoming, Laramie, Wyoming, United States of America, **2** Department of Molecular Biology and Biochemistry, Rutgers University, Piscataway, New Jersey, United States of America, **3** Department of Biochemistry and Molecular Biology, School of Basic Medicine, Tongji Medical College, Huazhong University of Science and Technology, Wuhan, Hubei, China

‡ These authors are joint first authors on this work.
* davidfay@uwyo.edu

**Data Availability Statement:** All relevant data are within the manuscript and its Supporting Information files. Raw data for all figures is contained in S1 File.

## Abstract

Endocytosis, the process by which cells internalize plasma membrane and associated cargo, is regulated extensively by posttranslational modifications. Previous studies suggested the potential involvement of scores of protein kinases in endocytic control, of which only a few have been validated in vivo. Here we show that the conserved NIMA-related kinases NEKL-2/NEK8/9 and NEKL-3/NEK6/7 (the NEKLs) control clathrin-mediated endocytosis in *C. elegans*. Loss of NEKL-2 or NEKL-3 activities leads to penetrant larval molting defects and to the abnormal localization of trafficking markers in arrested larvae. Using an auxin-based degron system, we also find that depletion of NEKLs in adult-stage *C. elegans* leads to gross clathrin mislocalization and to a dramatic reduction in clathrin mobility at the apical membrane. Using a non-biased genetic screen to identify suppressors of *nekl* molting defects, we identified several components and regulators of AP2, the major clathrin adapter complex acting at the plasma membrane. Strikingly, reduced AP2 activity rescues both *nekl* mutant molting defects as well as associated trafficking phenotypes, whereas increased levels of active AP2 exacerbate *nekl* defects. Moreover, in a unique example of mutual suppression, NEKL inhibition alleviates defects associated with reduced AP2 activity, attesting to the tight link between NEKL and AP2 functions. We also show that NEKLs are required for the clustering and internalization of membrane cargo required for molting. Notably, we find that human NEKs can rescue molting and trafficking defects in *nekl* mutant worms, suggesting that the control of intracellular trafficking is an evolutionarily conserved function of NEK family kinases.

## Author summary

In order to function properly, cells must continually import materials from the outside. This process, termed endocytosis, is necessary for the uptake of nutrients and for interpreting signals coming from the external environment or from within the body. These

**Funding:** This work was supported by the National Institutes of Health Institutes of General Medicine (NIGMS) GM066868 to DSF. This publication was also made possible by an Institutional Development Award (IDeA) from the National Institute of General Medical Sciences of the National Institutes of Health under Grant # 2P20GM103432, which partially supported BJ and VL. Some strains were provided by the Caenorhabditis Genetics Center (CGC), which is funded by the US National Institutes of Health (NIH) Office of Research Infrastructure Programs (P40 OD010440). The funders had no role in study design, data collection and analysis, decision to publish, or preparation of the manuscript.

**Competing interests:** The authors have declared that no competing interests exist.

signals are critical during animal development but also affect many types of cell behaviors throughout life. In our current work, we show that several highly conserved proteins in the nematode *Caenorhabditis elegans*, NEKL-2 and NEKL-3, regulate endocytosis. The human counterparts of NEKL-2 and NEKL-3 have been implicated in cardiovascular and renal diseases as well as many types of cancers. However, their specific functions within cells is incompletely understood and very little is known about their role in endocytosis or how this role might impact disease processes. Here we use several complementary approaches to characterize the specific functions of *C. elegans* NEKL-2 and NEKL-3 in endocytosis and show that their human counterparts likely have very similar functions. This work paves the way to a better understanding of fundamental biological processes and to determining the cellular functions of proteins connected to human diseases.

## Introduction

The cuticle of *C. elegans* is a flexible apical extracellular matrix consisting of cross-linked collagens, non-collagenous proteins, linked carbohydrates, and lipids [1, 2]. The cuticle is essential for providing a protective barrier from the environment, for maintaining the proper shape and integrity of the organism, and for facilitating muscle-based locomotion by functioning as an exoskeleton [3]. Remodeling of the cuticle occurs at the end of each of four larval stages (L1–L4) through a process called molting. During molting, a new cuticle is synthesized under the old cuticle, which is then shed [3–5]. Molting enables organismal growth and allows for changes in the composition and organization of the cuticle at different life stages. Molting defects can occur when synthesis of the new cuticle is compromised or when shedding of the old cuticle is incomplete. A sizeable number of factors have been implicated in *C. elegans* molting including proteins involved in cuticle structure, protein modification, protein degradation, cell signaling, transcription, and intracellular trafficking [4, 6].

During each molt, an accumulation of ribosomes, Golgi bodies and RNA is observed within the epidermal cells that produce the new cuticle, consistent with increased protein synthesis [1, 2, 5, 7]. The secretion of essential structural proteins, along with the enzymatic activities required for cuticle replacement, is accomplished through exocytosis. Consistent with this, inhibition of *sec-23*, which encodes a component of COPII-coated vesicles required for the transport of proteins from the endoplasmic reticulum (ER) to the Golgi, leads to molting defects [8]. At the same time, endocytosis is required to balance exocytosis and thus maintain a relatively constant volume/area of apical plasma membrane. In addition, the recycling of old cuticle components may be enabled through the process of endocytosis.

Endocytosis by the epidermis is also essential for the uptake of sterols from the environment, which provide building blocks for the hormonal cues that drive molting [4, 6, 9–12]. Consistent with this, worms deprived of cholesterol fail to molt [13–15]. Sterol uptake by the epidermis is thought to be dependent in part on LRP-1 (human LRP2), which belongs to the low-density lipoprotein (LDL) receptor family of integral membrane proteins. Inhibition of LRP-1 and other trafficking components required for LRP-1 uptake, such as the adapter protein DAB-1 (human DAB1/2), also lead to defective molting [15–19].

We previously reported that knockdown of NEKL-2 (human NEK8/9) or NEKL-3 (human NEK6/7), two conserved members of the Never-In-Mitosis-A (NIMA) protein kinase family, leads to molting defects in *C. elegans* [20, 21]. In addition, loss of function in the conserved ankyrin repeat proteins MLT-2 (human ANKS6), MLT-3 (human ANKS3), and MLT-4 (human INVS), leads to molting defects that are identical to those of *nekl* mutants. The

NEKL–MLTs form two distinct complexes (NEKL-2 with MLT-2–MLT-4 and NEKL-3 with MLT-3), and the MLTs are required for the correct subcellular localization of the NEKLs [21]. Importantly, these physical and functional interactions between NEKLs and MLTs appear to be highly conserved [22–25].

We previously showed that NEKL–MLTs are expressed in punctate patterns in the major syncytial epidermis of the worm, hyp7, suggesting that NEKL–MLTs localize to one or more trafficking compartments [20, 21]. In addition, we found that molting-defective *nekl–mlt* larvae exhibit abnormal morphology and/or localization of multiple trafficking markers in the apical region of hyp7, including a multi-copy clathrin heavy chain reporter [20, 21]. However, it was unclear as to whether the apparent defects in clathrin-mediated endocytosis were a primary cause of the observed molting defects in *nekl–mlt* mutants or a secondary consequence of physiological effects caused by the presence of a double cuticle and the inability of the worms to feed.

Clathrin-mediated endocytosis is a highly regulated stepwise process involving dozens of factors that act temporally to control the initiation, maturation, and internalization of clathrin-coated pits/vesicles [26–32]. Among these are the components of the clathrin scaffold itself, multiple triskelion units containing three heavy chains and an associated light chain. In addition, clathrin-mediated endocytosis requires numerous adapter proteins that link clathrin to the plasma membrane and to integral membrane cargo. Chief among these is the conserved plasma membrane adapter protein complex, AP2 [29, 33–35]. AP2 consists of four subunits, termed α, β, μ, and σ, and exists in at least two functionally distinct structures, broadly termed the open/active and closed/inactive conformations [27, 29, 36–39]. Allosteric regulators of AP2 conformation include FCHo1/2, which promotes the open state of AP2 [40, 41], and NECAP1/2, which promotes the closed conformation [42]. In addition, protein kinases have been implicated in the regulation of AP2 through phosphorylation of the μ subunit [39, 43].

Here we demonstrate that NEKL–MLTs regulate clathrin-mediated endocytosis within the context of an intact developing organism. We show that the function of NEKL–MLTs in trafficking is highly sensitive to the balance between the open and closed AP2 conformations and that AP2-associated phenotypes are also responsive to NEKL activity. In addition, we demonstrate that loss of NEKL functions leads to defects in LRP-1/LRP2 endocytosis, a cargo that is physiologically relevant to molting. Our combined findings indicate that defects in endocytosis are likely to be a major underlying basis for the observed molting defects in *nekl* mutants. Finally, we show that mammalian NEK6 and NEK7 can partially rescue endocytosis and molting defects in *nekl-3* mutant worms.

## Results

### *nekl–mlt* defects are suppressed by decreased function of the AP2 clathrin-adapter complex

We previously described a genetic and bioinformatic approach to identify suppressors of larval lethality in strains deficient for NEKL kinase activity [44]. Our screen makes use of weak aphenotypic alleles of *nekl-2(fd81)* and *nekl-3(gk894345)*, which, when combined in double mutants, lead to penetrant larval arrest due to molting defects [44, 45]. Homozygous *nekl-2(fd81); nekl-3 (gk894345)* mutants (hereafter referred to as *nekl-2; nekl-3* mutants) can be propagated only in the presence of a *nekl-2*⁺ or *nekl-3*⁺ rescuing extrachromosomal array, whereas strains that acquire a suppressor mutation no longer require the array for viability (Fig 1A and 1B).

Among *nekl-2; nekl-3* animals containing the *fd155* suppressor mutation, 98% progressed to adulthood versus only 1–2% for the parental *nekl-2; nekl-3* strain (Fig 1B and 1F). We showed the *fd155* causal mutation to be a T-to-A transversion in the 11th exon of *dpy-23*, which leads to

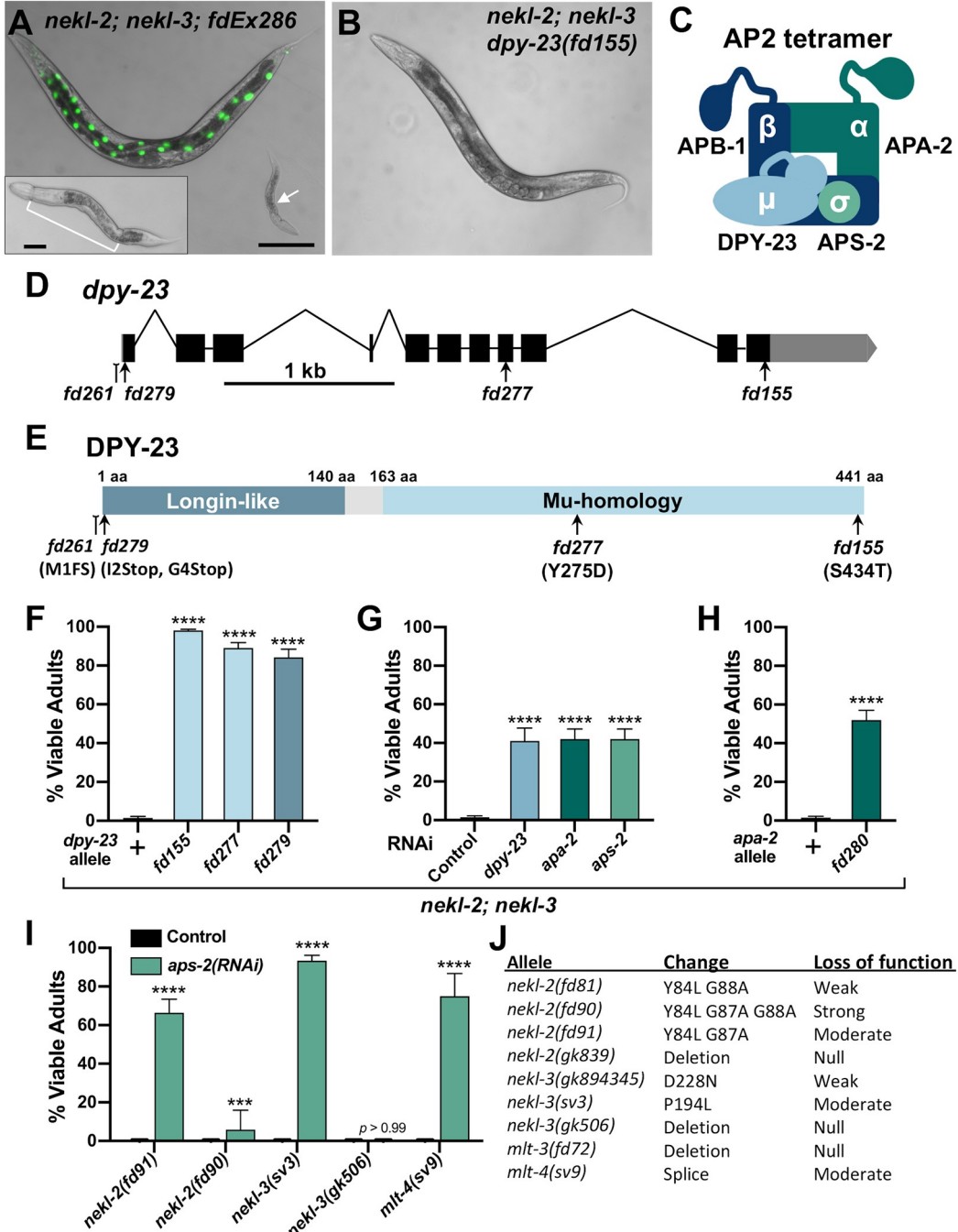

**Fig 1. Loss of AP2 complex activity suppresses *nekl–mlt* molting defects.** (A) Merged DIC and fluorescence images of *nekl-2(fd81); nekl-3(gk894345)* worms. The rescuing extrachromosomal array (*fdEx286*) expresses wild-type *nekl-3* and the *SUR-5::GFP* reporter. White arrow indicates a growth-arrested *nekl-2; nekl-3* larva that failed to inherit the array and exhibits a "corset" morphology, characteristic of *nekl–mlt* molting defects (inset in A). The bracket marks the constricted mid-body region, which contains a double cuticle. Bar size in A = 100 μm (for A and B); in inset, 20 μm. (B) DIC image of *nekl-2; nekl-3* adult worm containing the suppressor mutation *dpy-23(fd155)*. (C) Graphic representation of AP2 tetramer complex containing four subunits. (D) Gene structure diagram of *dpy-23/*μ including the locations of mutations. Point mutations (*fd155*, *fd277*, *fd279*) are indicated by arrows; indel (*fd261*) by the line ending in a small bracket. (E) Protein domain diagram of DPY-23/μ with corresponding allelic changes; *fd261* is missing ~30 bp of the proximal 5'UTR including a predicted SL1 transplice site and the start codon. The amino acid (aa) locations of the two domains are indicated. (F–I) Bar plots showing suppression of molting defects in *nekl–mlt* mutants by reduction in AP2 activity. Assays F–H were carried out in *nekl-2(fd81); nekl-3(gk894345)* double mutants. (G,I) RNAi was carried out in the indicated backgrounds

using injection methods; control indicates non-injected siblings. (F–I) Error bars indicate 95% confidence intervals; p–values were determined using Fischer's exact test where proportions were compared to the wild-type allele (F,H) or to the RNAi control (G,I): ****$p < 0.0001$, ***$p < 0.001$. (J) Guide to *nekl–mlt* alleles used in this study. Raw data are available in S1 File.

a S434T substitution in a serine that is highly conserved (Fig 1D and 1E). A second independent mutation (*fd277*) was identified as a T to G transversion in the 8th exon of *dpy-23*, leading to a Y275D substitution in a highly conserved tyrosine residue (Fig 1D and 1E). Both mutations, as well as a CRISPR/Cas9-generated predicted null allele of *dpy-23* (*fd279*) (S1 Table), led to a high percentage of viable adults in the *nekl-2; nekl-3* background (Fig 1D–1F), and partial suppression was observed in *nekl-2; nekl-3* strains treated with *dpy-23(RNAi)* (Fig 1G).

*dpy-23* (also known as *apm-2*) encodes the μ subunit (also termed μ2) of the *C. elegans* AP2 complex [46, 47]; the other three subunits are encoded by *apa-2* (α), *apb-1* (β), and *aps-2* (σ) (Fig 1C) [29, 48]. AP2 binds to phosphatidylinositol-4,5-bisphosphate (PIP2) lipids on the plasma membrane and functions as an adapter, linking cytoplasmic clathrin to plasma membrane cargo [29, 48]. In *C. elegans*, AP2 subunits form two partially independent hemicomplexes composed of μ/β and α/σ [49]. Although normal levels of clathrin-mediated endocytosis occur only when all four subunits are functional, *C. elegans* strains containing either the μ/β or α/σ hemicomplex are nevertheless viable, as demonstrated by the ability to propagate homozygous null mutants of *apa-2*/α, *dpy-23*/μ, or *aps-2*/σ [46, 47, 49]. Strains containing strong loss of function in *apb-1*, however, are embryonic lethal because of the role of APB-1/β1/2 in both AP2 and AP1 complexes, the latter of which function in clathrin-mediated trafficking from the trans-Golgi network and from endosomes [35, 49–51].

To determine if depletion of the other AP2 subunits could suppress *nekl* molting defects, we carried out RNAi in *nekl-2; nekl-3* animals. RNAi of *apa-2*/α or *aps-2*/σ allowed ~40% of *nekl-2; nekl-3* animals to reach adulthood (Fig 1G). Furthermore, a CRISPR-generated null allele of *apa-2*/α (*fd280*) led to a slightly higher proportion of *nekl-2; nekl-3* animals reaching the adult stage (Fig 1H, S1 Table). Attempts to suppress *nekl-2; nekl-3* arrest by *apb-1(RNAi)* were unsuccessful due to early embryonic lethality, as expected. Our findings indicate that suppression of *nekl-2; nekl-3* molting defects are not specific to individual AP2 subunits or hemicomplexes, although loss of the μ/β hemicomplex may provide a higher level of suppression than loss of α/σ (Fig 1F and 1H).

We next determined if suppression of *nekl-2; nekl-3* double mutants by inhibition of AP2 was specific to either the NEKL-2 or NEKL-3 pathways. For these tests, we first carried out *aps-2(RNAi)* in *nekl–mlt* backgrounds containing moderate-to-strong loss-of-function alleles (Fig 1I and 1J) that exhibit 100% larval arrest as single mutants [20, 45]. Downregulation of *aps-2*/σ led to significant suppression of larval lethality in moderate-to-strong loss-of-function alleles of *nekl-2*, *nekl-3*, and *mlt-4* (Fig 1I), with weakest suppression observed for *fd90*, a relatively strong *nekl-2* allele. In contrast, no suppression was observed in strains containing a null allele of *nekl-3*(*gk506*) (Fig 1I).

Failure to observe suppression of *nekl–mlt* null alleles using RNAi to knockdown AP2 may be due to incomplete inactivation of the targeted subunit. We therefore determined if null alleles in *apa-2*/α or *dpy-23*/μ could suppress larval arrest in null *nekl–mlt* backgrounds. In the case of *apa-2*/α, we failed to observe any suppression in null *nekl-2*, *nekl-3*, and *mlt-3* backgrounds ($n > 1000$ for each strain), indicating that complete loss of the α/σ hemicomplex is not sufficient to overcome the requirement for NEKL-2 and NEKL-3. Attempts to score suppression of *nekl–mlt* null alleles with *dpy-23(e840)*, however, were not successful because the generated compound mutants were very sick and slow growing.

## Loss of FCHO-1, an activator of AP2, suppresses *nekl–mlt* defects

Among *nekl-2; nekl-3* animals containing the *fd131* suppressor mutation, 87% progressed to adulthood (Fig 2A and 2E). Prior genetic characterization of *fd131* indicated the causal mutation to be recessive and autosomal [44]. We determined the *fd131* causal mutation to be a C-to-T transition in the 13th exon of *fcho-1*, which introduces a premature stop codon following amino acid (aa) 903 (Q904Stop; Fig 2C and 2D, S1 Table). This mutation is predicted to truncate the 963-aa FCHO-1 protein within the conserved Mu-Homology domain (Fig 2C and 2D). Consistent with this, CRISPR-generated truncations within the Mu-Homology domain (*fd211* and *fd212*) led to a high percentage of viable adults in the *nekl-2; nekl-3* background, as did a null deletion allele of *fcho-1* (*ox477*; Fig 2C–2E;; S1 Table). In addition, partial knockdown of *fcho-1* by RNAi resulted in ~40% of *nekl-2; nekl*-3 mutants reaching adulthood (Fig 2F).

FCHO-1 is a member of the muniscin protein family, members of which have important roles in clathrin-mediated endocytosis and are recruited to nascent pits early during endocytic vesicle formation [27, 32, 52, 53]. Orthologs of FCHO-1 contain three characterized functional domains: an N-terminal F-BAR domain, a C-terminal Mu-homology domain, and an internal AP2 Activator domain (Fig 1D). The F-BAR domain binds to lipid bilayers and aids in membrane curvature, whereas the Mu-homology domain binds to cargo and other endocytic adapter proteins. The AP2 Activator domain has recently been shown to facilitate the allosteric opening/activation of AP2 (Fig 2B), and loss of *fcho-1* activity in *C. elegans* leads to a decrease in the levels of open/active AP2 [39–41].

Similar to AP2 subunits, suppression of *nekl–mlt*s by *fcho-1(RNAi)* was not specific to either the NEKL-2 or NEKL-3 pathways (Fig 2F). Also like the AP2 subunits, *fcho-1(RNAi)* failed to suppress strong loss-of-function alleles of *nekl-2* and *nekl-3* (Fig 2F). Interestingly, unlike AP2 subunits, a null deletion allele of *fcho-1* (*ox477*) was able to suppress strong/null alleles of *nekl-2, nekl-3,* and *mlt-3* (Fig 2G).

The above findings imply that decreasing the amount of open/active AP2, either by reducing gross AP2 levels or by inhibiting its allosteric activation, can bypass the requirement for NEKL–MLT activity. As a further test of this model, we used CRISPR to generate an in-frame *fcho-1* deletion (*fd262*), which is predicted to specifically remove the AP2 activator domain of FCHO-1 without affecting the F-BAR or Mu-Homology domains (Fig 2C and 2D) [40, 41]. Importantly, *fcho-1(fd262)* strongly suppressed molting defects in *nekl-2; nekl-3* mutants (Fig 2E), consistent with suppression by *fcho-1* occurring through a reduction in AP2 activity.

## Excess open AP2 enhances *nekl–mlt* molting defects

Given that decreasing the amount of open AP2 led to the suppression of *nekl–mlt* defects, we next tested if increasing the levels of open AP2 could enhance molting defects in weak *nekl–mlt* loss-of-function backgrounds. *ncap-1* encodes the *C. elegans* ortholog of the mammalian adaptiN Ear-binding Coat-Associated Proteins (NECAP1 and NECAP2), which function in several different aspects of clathrin-mediated endocytosis [42, 54–57]. Recently, *C. elegans* NCAP-1 was shown to allosterically regulate AP2 to promote the closed/inactive conformation, and loss of *ncap-1* suppresses the slow-growth phenotype of *fcho-1* strains [42]. Thus, NCAP-1 acts in opposition to FCHO-1 to regulate AP2 conformation and activity (Fig 3A) [39].

To test for enhancement of molting defects, we carried out RNAi of *nekl–mlt*s in wild-type and *ncap-1(mew39)* deletion backgrounds using "weak" RNAi feeding methods, which cause only a partial reduction in *nekl–mlt* activities [20, 21]. Whereas RNAi feeding of *nekl–mlt*s in wild type resulted in little or no defective molting, significantly elevated levels of molting arrest were detected in the *ncap-1(mew39)* background (Fig 3B). These findings imply that increased levels of open AP2 exacerbate *nekl–mlt* loss-of-function phenotypes. Correspondingly, RNAi

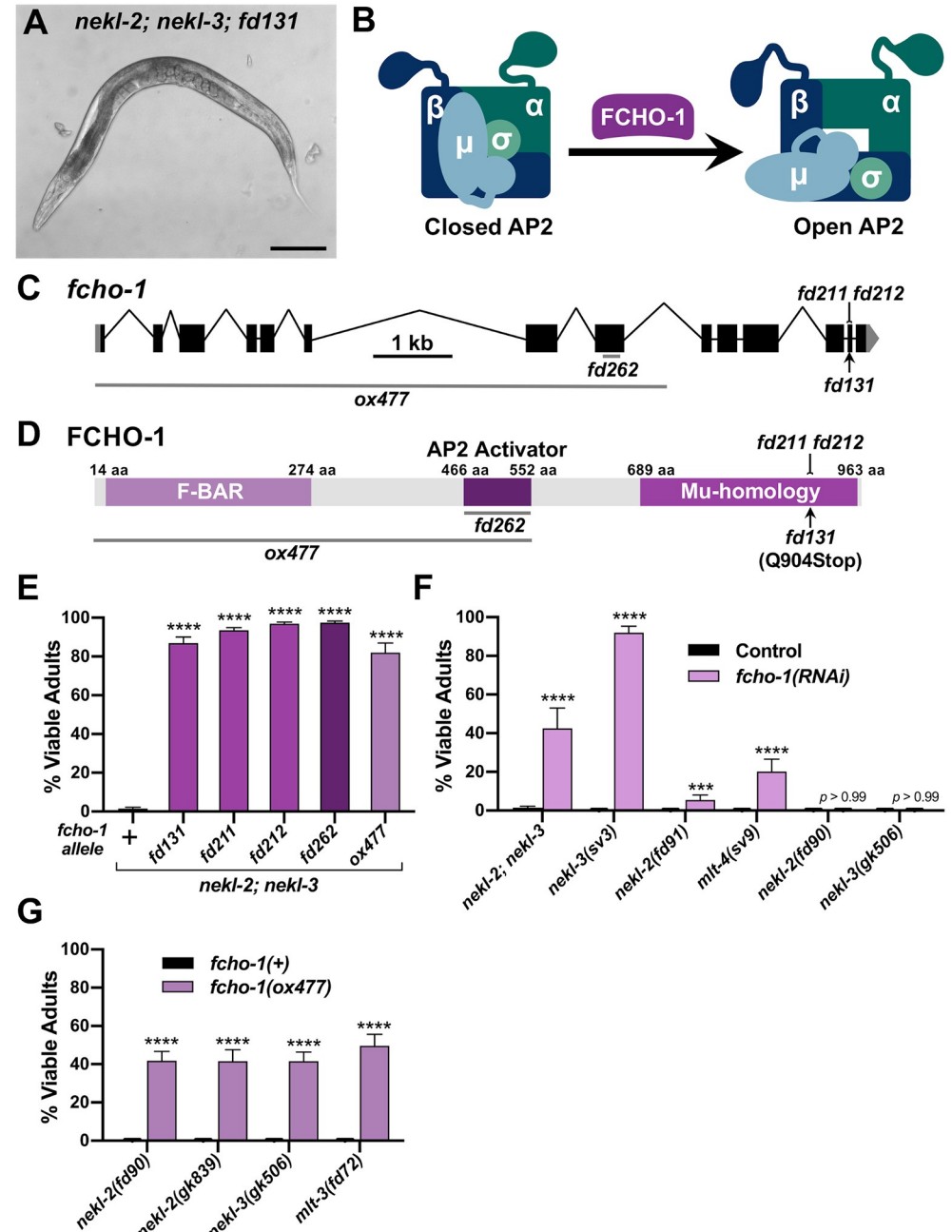

**Fig 2. Loss of *fcho-1* activity suppresses *nekl–mlt* molting defects.** (A) DIC image of *nekl-2(fd81); nekl-3(gk894345)* adult worm containing the suppressor mutation *fd131*. Bar size in A = 100 μm. (B) Model of AP2 allosteric regulation by FCHO-1. (C) Gene structure diagram of *fcho-1* including the locations of mutations. The point mutation *fd131* is indicated by the arrow, indels (*fd211*, *fd212*) by the line ending in a small bracket, and deletions (*ox477*, *fd262*) by horizontal gray lines. (D) Protein domain diagram of FCHO-1 with corresponding allelic changes. The amino acid (aa) locations of the three domains are indicated. (E–G) Bar plots showing suppression of molting defects in *nekl–mlt* mutants by reduction of FCHO-1 activity. (E) Assays were carried out in *nekl-2(fd81); nekl-3(gk894345)* double mutants using the indicated *fcho-1* alleles. (F) RNAi was carried out in the indicated backgrounds using injection methods; control indicates non-injected siblings. (G) Assays were carried out in strong/null *nekl–mlt* backgrounds using the null *fcho-1(ox477)* allele. (E–G) Error bars indicate 95% confidence intervals. p–Values were determined using Fischer's exact test where proportions were compared to the wild-type allele (E,G) or to the RNAi control (F); ****$p < 0.0001$, ***$p < 0.001$. Raw data are available in S1 File.

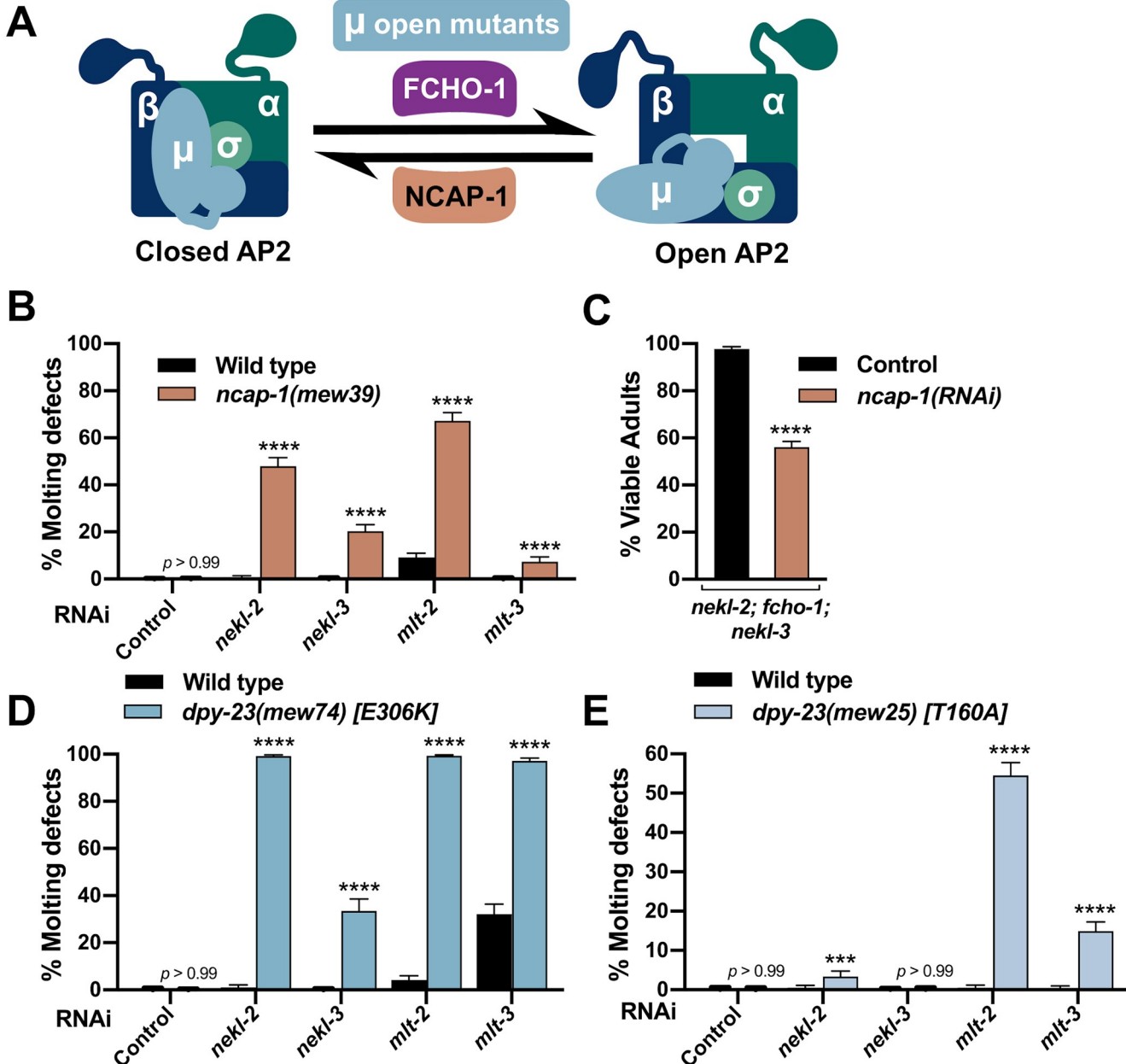

**Fig 3. Excess open AP2 enhances *nekl–mlt* molting defects.** (A) Model depicting the allosteric regulation of AP2 by FCHO-1 and NCAP-1. DPY-23/μ open mutants cause AP2 to remain in the open/active state. (B,D,E) Bar plots showing enhancement of molting defects by mutations in *ncap-1* (B) and *dpy-23*/μ open mutants (D,E). RNAi feeding was carried out for the indicated genes; control RNAi feeding targeted GFP. (C) Bar plot showing partial reversion of suppression in *nekl-2(fd81); fcho-1(fd131); nekl-3(gk894345)* triple mutants after *ncap-1(RNAi)* was carried out using injection methods. Control indicates non-injected siblings. (B–E) Error bars indicate 95% confidence intervals. p-Values were determined using Fischer's exact test where proportions were compared to the RNAi controls; ****$p < 0.0001$, ***$p < 0.001$. Raw data are available in S1 File.

of *ncap-1* mitigated suppression conferred by loss of *fcho-1* in *nekl-2; nekl-3* mutants (Fig 3C). These results are consistent with the reported opposing functions for NCAP-1 and FCHO-1 and support the model that suppression of *nekl–mlts* by *fcho-1* and AP2 mutations is due to reduced levels of open AP2.

Because mammalian NECAPs have been suggested to have several distinct functions during endocytosis [54–56, 58], we carried out additional tests to determine if increased open AP2

could enhance *nekl–mlt* defects. For these studies, we made use of several missense mutations in *dpy-23*/μ that shift the balance of AP2 toward the open state ("open mutants") [40]. As with *ncap-1*, we observed strong enhancement of molting defects after *nekl–mlt* RNAi feeding in *dpy-23*/μ strains containing E306K or T160A substitutions (Fig 3D–3E). The somewhat stronger findings observed for the E306K mutation correlates with biochemical assays showing that E306K leads to higher levels of open AP2 than the T160A substitution [40]. Collectively, our findings demonstrate that reducing the level of open AP2 suppresses *nekl–mlt* molting defects whereas increasing the level of open AP2 exacerbates defects.

## Loss of NEKL–MLT activity suppresses AP2-associated defects

Loss of function of non-essential AP2 subunits (μ, α, and σ) and *fcho-1* leads to reduced growth rates and the accumulation of fluid between the cuticle and epidermis [40, 49]. This latter defect visibly manifests in adult-stage worms as bilateral bulges located near the junction of the hyp6 and hyp7 epidermal syncytia in the region of the neck (the "Jowls" phenotype; Fig 4A). Strikingly, we observed that loss of function in NEKL–MLT activity significantly suppressed the Jowls phenotype of *fcho-1* and AP2 mutants (Fig 4A–4F). Specifically, partial suppression of Jowls was observed in the *fcho-1(ox477)* null mutant background, as well as in the presence of null alleles of *apa-2*/α and *dpy-23*/μ, and in *aps-2(RNAi)* animals (Fig 4C–4F). Moreover, suppression was observed in strains with reduced or abolished function in either the NEKL-2 or NEKL-3 pathways (Fig 4C and 4F). Our observation that simultaneous loss of AP2 and NEKL–MLT activities leads to the mutual suppression of both molting-defective and Jowls phenotypes underscores the tight functional connection between NEKL–MLT and AP2 activities in vivo.

## Molting-defective *nekl* mutants exhibit changes in epidermal clathrin that are suppressed by inhibition of *fcho-1* and AP2

Given the strong genetic data linking the NEKL–MLTs to AP2, we next examined the role of NEKL–MLTs in clathrin-mediated endocytosis. We previously showed that molting-defective *nekl–mlt* mutants exhibit abnormal localization of a multi-copy clathrin heavy chain reporter [20, 21, 59]. To visualize clathrin at physiological levels, we used CRISPR/Cas9 to generate a clathrin heavy chain reporter construct with an N-terminal GFP tag (GFP::CHC-1), which showed the expected punctate localization pattern in the hyp7 epidermal syncytium (Fig 5A). Consistent with our previous findings, GFP::CHC-1 localization was altered in molting-defective *nekl-2; nekl-3* larvae (Fig 5A and 5B). Specifically, the average mean intensity of GFP::CHC-1 was increased by 2.2-fold in the apical region of hyp7 in *nekl-2; nekl-3* larvae arrested at the L2/L3 molt relative to wild-type late-stage L2 larvae (Fig 5B). In addition, the percentage of GFP-positive pixels (above a uniformly applied threshold) was increased by 1.3-fold in *nekl-2; nekl-3* molting-defective larvae relative to wild type, which may reflect an increase in the density and/or size of apical GFP::CHC-1 puncta (Fig 5A–5C).

We next determined if clathrin defects in *nekl-2; nekl-3* larvae could be suppressed by a reduction in FCHO-1 and AP2 activities. Relative to *nekl-2; nekl-3* mutants, the average mean intensity of apical GFP::CHC-1 was reduced by 1.5-fold in *nekl-2; fcho-1(fd131); nekl-3* late-stage L2 larvae, and the percentage of positive pixels was reduced by 1.4-fold (Fig 5A–5C). A similar trend was observed for *nekl-2; dpy-23(fd155) nekl-3* animals, which exhibited a 1.2-fold decrease in both the GFP::CHC-1 mean intensity and in the percentage of pixels above threshold relative to *nekl-2; nekl-3* animals, although the observed change in mean intensity was not statistically significant (Fig 5A–5C). Together, our findings suggest that reduced AP2 activity mitigates both molting and clathrin-localization defects in *nekl-2; nekl-3* larvae.

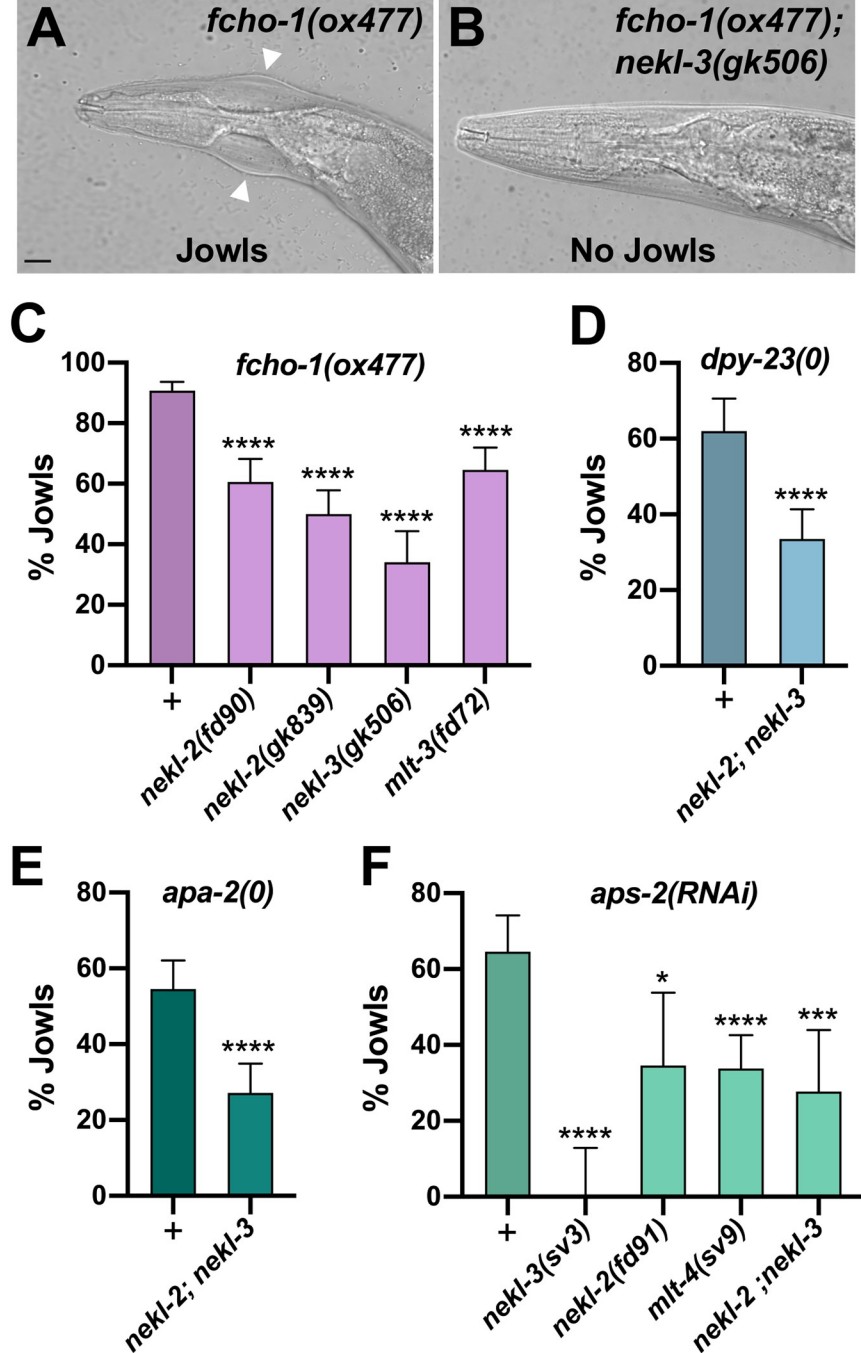

**Fig 4. The AP2-associated Jowls phenotype is suppressed by *nekl–mlt* mutants.** (A,B) Representative DIC images of *fcho-1(ox477)* (A) and *fcho-1(ox477); nekl-3(gk506)* (B) adults. White arrowheads mark the location of Jowls in *fcho-1 (ox477)* mutants. Bar size in A = 5 μm (for A and B). (C–F) The percentage of adult animals exhibiting the Jowls phenotype was assayed in the indicated genetic backgrounds; "+" indicates that no *nekl–mlt* mutations were present. (C) Assays were carried out in strong/null *nekl–mlt* backgrounds using the null *fcho-1(ox477)* allele. (D) Assays were carried out using the *dpy-23*/μ null alleles *fd261* (in the + background), and *fd279* (in the *nekl-2(fd81); nekl-3 (gk894345)* background). (E) Assays were carried out using the *apa-2* null alleles *fd282* (in the + background) and *fd280* (in the *nekl-2(fd81); nekl-3(gk894345)* background). (F) RNAi was carried out in the indicated backgrounds using injection methods. Error bars indicate 95% confidence intervals. p-Values were determined using Fischer's exact test where proportions were compared to the corresponding wild-type *nekl–mlt* allele (+) (C,D,E) or non-injected controls (F); $^{****}p < 0.0001$, $^{***}p < 0.001$, $^{*}p < 0.05$. Raw data are available in S1 File.

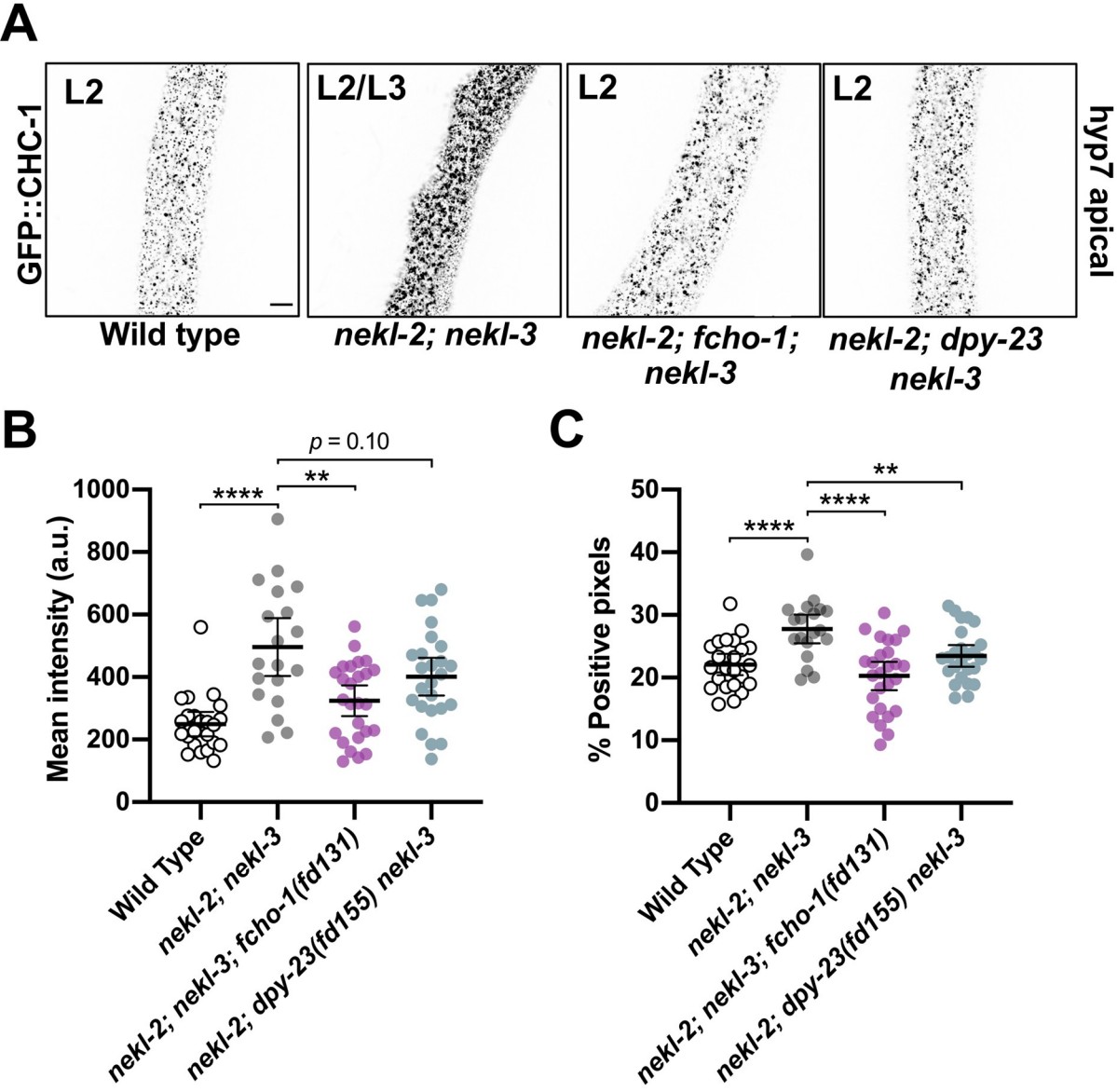

**Fig 5. Clathrin defects in *nekl-2; nekl-3* mutants are suppressed by reduced AP2 activity.** (A) Representative confocal images of L2–L3 larvae expressing CRISPR-tagged GFP::CHC-1 within the apical region of the hyp7 epidermal syncytium. GFP::CHC-1 localization is shown for wild-type, *nekl-2(fd81); nekl-3(gk894345)*, *nekl-2(fd81); fcho-1(fd131); nekl-3(gk894345)*, and *nekl-2(fd81); dpy-23(fd155) nekl-3(gk894345)* strains. Inverted fluorescence images are shown to aid clarity. Bar size in A = 5 μm (for A–D). Background subtraction was performed using the same parameters for all images; minimum and maximum pixel values were kept consistent for all images. (B,C) For individual larvae of the indicated genotypes, the mean GFP::CHC-1 intensity (B) and the percentage of GFP-positive pixels above threshold (C) were determined in the apical region of hyp7. Both the group mean and 95% confidence interval (error bars) are shown. p-Values for compared means were determined using two-tailed Mann-Whitney tests; ****$p < 0.0001$, **$p < 0.01$. Raw data are available in S1 File.

### Loss of NEKLs in adults leads to changes in clathrin that are independent of molting defects

Although our above findings suggest that the NEKLs regulate epidermal clathrin, the analysis of trafficking in *nekl–mlt* mutants is problematic because of potential secondary effects caused by the double cuticle and larval arrest phenotype. To circumvent this obstacle, we engineered regulatable NEKL kinases using the auxin-induced degradation system [60, 61]. Importantly, by depleting NEKLs after the final molt, we eliminated the possibility that changes in clathrin

localization could be an indirect consequence of defective molting. Proteins tagged with an auxin-inducible degron (AID) are responsive to auxin, which binds to the AID motif leading to ubiquitination by the TIR1–SCF E3-ligase complex and degradation by the proteosome (Fig 6A).

Exposure of CRISPR-tagged NEKL-2::AID and NEKL-3::AID day-1 adults to auxin led to the complete loss of both full-length tagged proteins within 20 h (Fig 6B). Consistent with NEKL loss of function, 100% of NEKL-2::AID and NEKL-3::AID L1 larvae exposed to auxin from hatching arrested with molting defects (Fig 6C) and displayed abnormal accumulation of apical GFP::CHC-1 (S1 Fig). We note that whereas auxin treatment resulted in the complete disappearance of any detectable NEKL-3::AID protein, we often observed a more rapidly migrating band in samples from auxin-treated NEKL-2::AID worms, suggesting that a partial fragment of NEKL-2::AID may be resistant to further degradation.

To determine the effects of auxin-induced NEKL::AID depletion on clathrin localization in adults, we exposed day-1 adults to auxin for 20 h prior to GFP::CHC-1 localization analysis. In control experiments with wild-type adults, auxin treatment alone did not significantly affect apical hyp7 GFP::CHC-1 localization (Fig 6J and 6K, S2 Fig; S1 and S2 Movies). Strikingly, we observed 2.5-fold and 3.6-fold increases in the average mean intensities of apical GFP::CHC-1 in auxin-treated NEKL-2::AID and NEKL-3::AID adults, respectively, relative to auxin-treated wild-type animals (Fig 6D, 6G and 6J; S3 and S4 Movies). Likewise, auxin-treated NEKL-2:: AID and NEKL-3::AID adults displayed 1.4-fold and 1.7-fold increases, respectively, in the percentage of pixels above threshold relative to auxin-treated wild-type animals (Fig 6D, 6G and 6K). These data demonstrate that loss of NEKLs leads to abnormal apical clathrin localization through a mechanism that is independent of the molting cycle. The stronger effects observed for NEKL-3 could be due to an incomplete loss of NEKL-2 activity after auxin treatment (Fig 6B) or to different requirements for NEKL-2 and NEKL-3 in clathrin-mediated endocytosis.

While conducting these studies, we also observed a clear effect of the AID tag on the activities of NEKL-2 and NEKL-3 even in the absence of auxin. Both mean GFP::CHC-1 levels and the percentage of pixels above threshold were increased in the apical hyp7 region of untreated NEKL-2::AID and NEKL-3::AID adults relative to untreated wild-type controls (Fig 6J and 6K; S5 and S6 Movies). The effect of the AID tag was strongest in the NEKL-3::AID strain, consistent with our observation that ~20% of untreated NEKL-3::AID worms displayed molting defects (Fig 6C, 6J and 6K). Higher baseline levels in the untreated NEKL::AID strains resulted in somewhat less-dramatic fold changes in comparisons of untreated versus auxin-treated NEKL::AID adults. For example, the average intensity of GFP::CHC-1 increased by 1.6-fold and 1.8-fold in NEKL-2::AID and NEKL-3::AID auxin-treated strains, respectively, compared with age-matched untreated NEKL::AID controls (Fig 6E and 6H). Correspondingly, the percentage of pixels above threshold was increased by 1.2-fold and 1.3-fold in auxin-treated NEKL-2::AID and NEKL-3::AID adults, respectively, relative to untreated controls (Fig 6, 6F and 6I). The observed effects in untreated NEKL::AID strains could be due to the 45-aa AID tag sterically interfering with activities of the NEKL proteins. Alternatively, some auxin-independent activity of the TIR1–E3 ubiquitin-ligase complex could lead to reduced levels of NEKL::AID proteins relative to wild type.

## NEKL depletion preferentially affects apical clathrin localization

To examine if NEKL depletion principally affects apical clathrin localization, we analyzed GFP::CHC-1 localization in medial planes of hyp7. Consistent with NEKLs having a primary effect on apical clathrin, NEKL depletion led to only modest changes in GFP::CHC-1

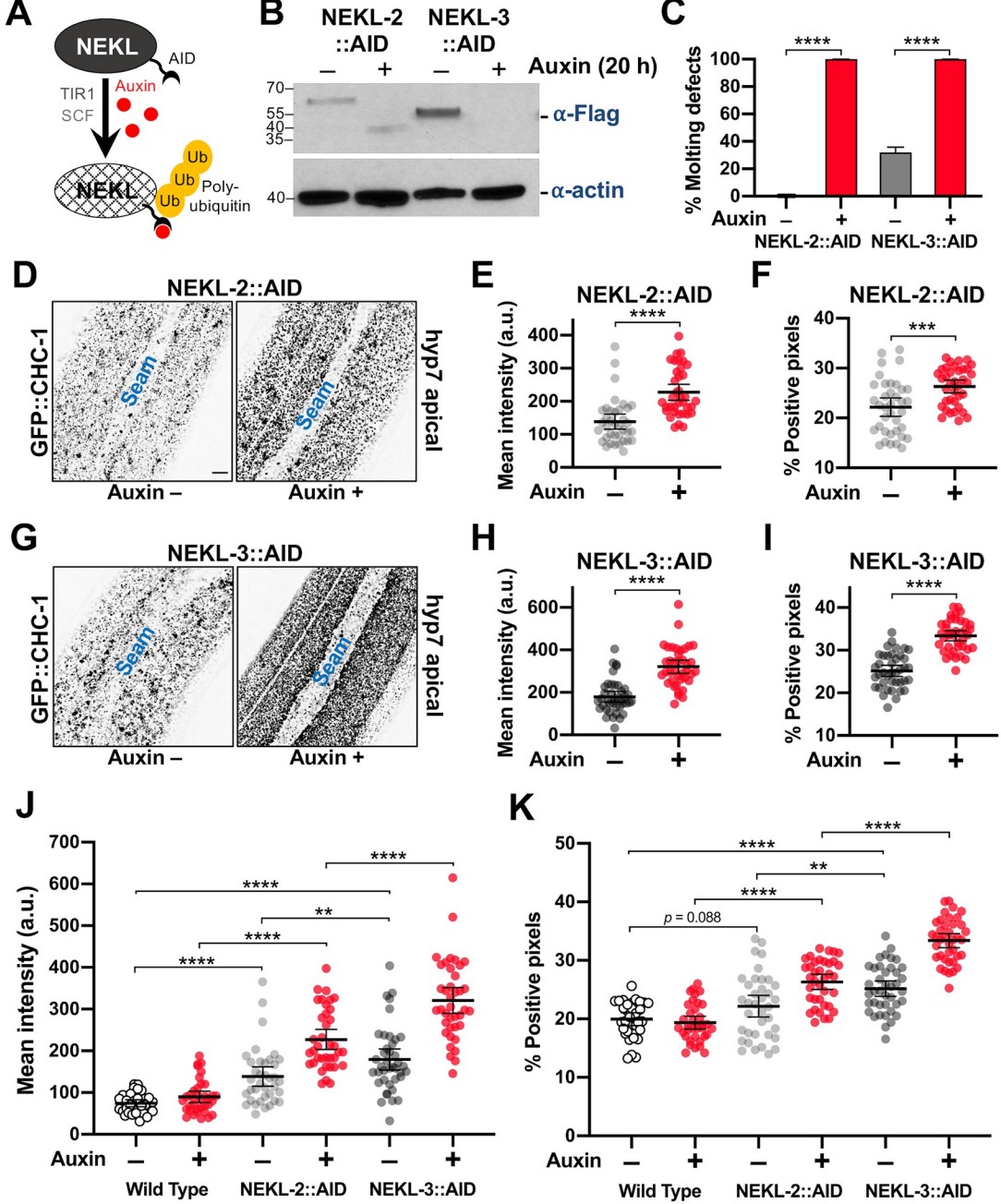

**Fig 6. Clathrin localization is altered in NEKL-depleted adults.** (A) Model of auxin-inducible degradation of NEKL kinases containing the auxin inducible degron (NEKL-2::AID and NEKL-3::AID). The addition of auxin triggers ubiquitination of NEKL::AID proteins followed by proteolysis. (B) Western blot showing complete loss of the full-length NEKL-2::AID and NEKL-3::AID proteins after auxin treatment. (C) Bar plot showing the percentage of molting defects among NEKL-2::AID and NEKL-3::AID strains in the presence and absence of auxin. p-Values were determined using Fischer's exact test; ****$p < 0.0001$. (D–K) CRISPR-tagged GFP::CHC-1 localization was analyzed in NEKL-2::AID and NEKL-3::AID strains in the presence and absence of auxin (20 h) in day-2 adults. (D,G) Representative confocal images of day-2 adults expressing GFP::CHC-1 within the apical region of the hyp7 epidermal syncytium. Background subtraction was performed using the same parameters for all images; minimum and maximum pixel values were kept consistent for all images. Inverted fluorescence was used to aid clarity. Bar size in D = 5 μm (for D and G). (E,F,H,I) Mean GFP::CHC-1 intensities (E,H) and the percentage of GFP-positive pixels above threshold (F,I) were determined for individual adults. (J,K) Summary comparison of data from panels E,F,H,I and S2 Fig (B,C). (E,F,H–K) The group mean and 95% confidence interval (error bars) are shown. p-Values were determined using two-tailed Mann-Whitney tests; ****$p < 0.0001$, ***$p < 0.001$, **$p < 0.01$. Raw data are available in S1 File.

localization in more medial planes, ~1.2 μm from the apical membrane (S2 Fig; S1–S6 Movies). Specifically, GFP::CHC-1 average mean intensity increased 1.2-fold in auxin-treated NEKL-2::AID and NEKL-3::AID adults relative to untreated NEKL::AID worms, and no significant differences were detected in the percentage of pixels above threshold (S2 Fig).

As an additional test, we simultaneously examined clathrin (GFP::CHC-1) and a marker for AP2, P$_{hyp7}$::mScarlet::DPY-23/μ2, in auxin-treated wild-type and NEKL-3::AID adults. Notably, the mean intensity and percentage of positive pixels increased for both clathrin and DPY-23/μ2 in NEKL-3::AID worms relative to wild type (Fig 7A–7I). The greater increase in mean intensity observed for clathrin versus DPY-23/μ2 may reflect the redistribution of clathrin to the apical surface from other compartments within the cell (Fig 7H). In contrast, the large majority of DPY-23/μ2 is expected to reside close the apical surface in wild type. We also note that the Pearson's r coefficient, which is a measure of CHC-1–DPY-23/μ2 colocalization, was significantly greater in auxin-treated NEKL-3::AID animals versus wild type (Fig 7G), also consistent with an increase in apical clathrin. Conversely, we observed a slight decrease in the extent of co-localization between clathrin and a marker for AP1, P$_{hpy7}$::mScarlet::APM-1/μ1, a subunit of the clathrin adapter complex acting in the trans-Golgi compartment (S3 Fig). Altogether our findings indicate that loss of NEKLs leads to increased accumulation of apical clathrin, reflecting increased recruitment or retention of clathrin at or near the plasma membrane.

## Depletion of NEKLs greatly reduces clathrin exchange on apical membranes of the epidermis

Our results above show overaccumulation of clathrin on apical hyp7 epidermal membranes after depletion of NEKL-2 or NEKL-3. Such an overaccumulation could be due to a higher rate of coated pit formation, a reduced rate of coated pit release from the plasma membrane, or a reduced rate of clathrin-coated vesicle uncoating. One way to differentiate among these possibilities is to measure clathrin exchange between the membrane-associated and cytoplasmic pools via fluorescence recovery after photobleaching (FRAP) assays using GFP::CHC-1. If coated pit assembly rates were increased in animals lacking NEKLs we would expect to observe increased recovery rates for GFP::CHC-1 after photobleaching. Conversely, if NEKL depletion leads to a decrease in the disassembly of coated vesicles, as occurs after auxilin depletion, we would expect to observe a decrease in recovery after photobleaching [62]. In addition, certain perturbations that block endocytosis *in vivo* (e.g., exposure to hypertonic sucrose media or intracellular potassium depletion) lead to the genesis of abnormal "clathrin microcages" just below the surface of the plasma membrane, which also block recovery from photobleaching [63]. Moreover, perturbation of the scission enzyme dynamin, which results in failed pinching off of clathrin-coated vesicles, does not affect GFP-clathrin recovery from photobleaching, even though coated pits over-accumulate on the plasma membrane [63]. Therefore, no change in recovery after photobleaching could indicate a role for the NEKLs in vesicle scission.

To determine if the NEKLs affect clathrin dynamics, we carried out FRAP in NEKL-2::AID and NEKL-3::AID strains ~20 h after exposure to auxin. In the case of wild-type controls, the mobile fraction of GFP::CHC-1 in the apical hyp7 region appeared to be slightly higher in untreated (75%) versus auxin-treated (64%) animals, although this difference was at most marginally significant (p = 0.071; Fig 8A, S4 Fig, S7 and S8 Movies). Strikingly, the GFP::CHC-1 mobile fraction in NEKL-3::AID adults was reduced to only 20% in auxin-treated worms, a more than 3-fold reduction relative to auxin-treated wild-type animals (Fig 8E, 8F and 8H, S4 Fig, S9 Movie). Similarly, the mobile fraction in NEKL-2::AID strains was just 44% in auxin-

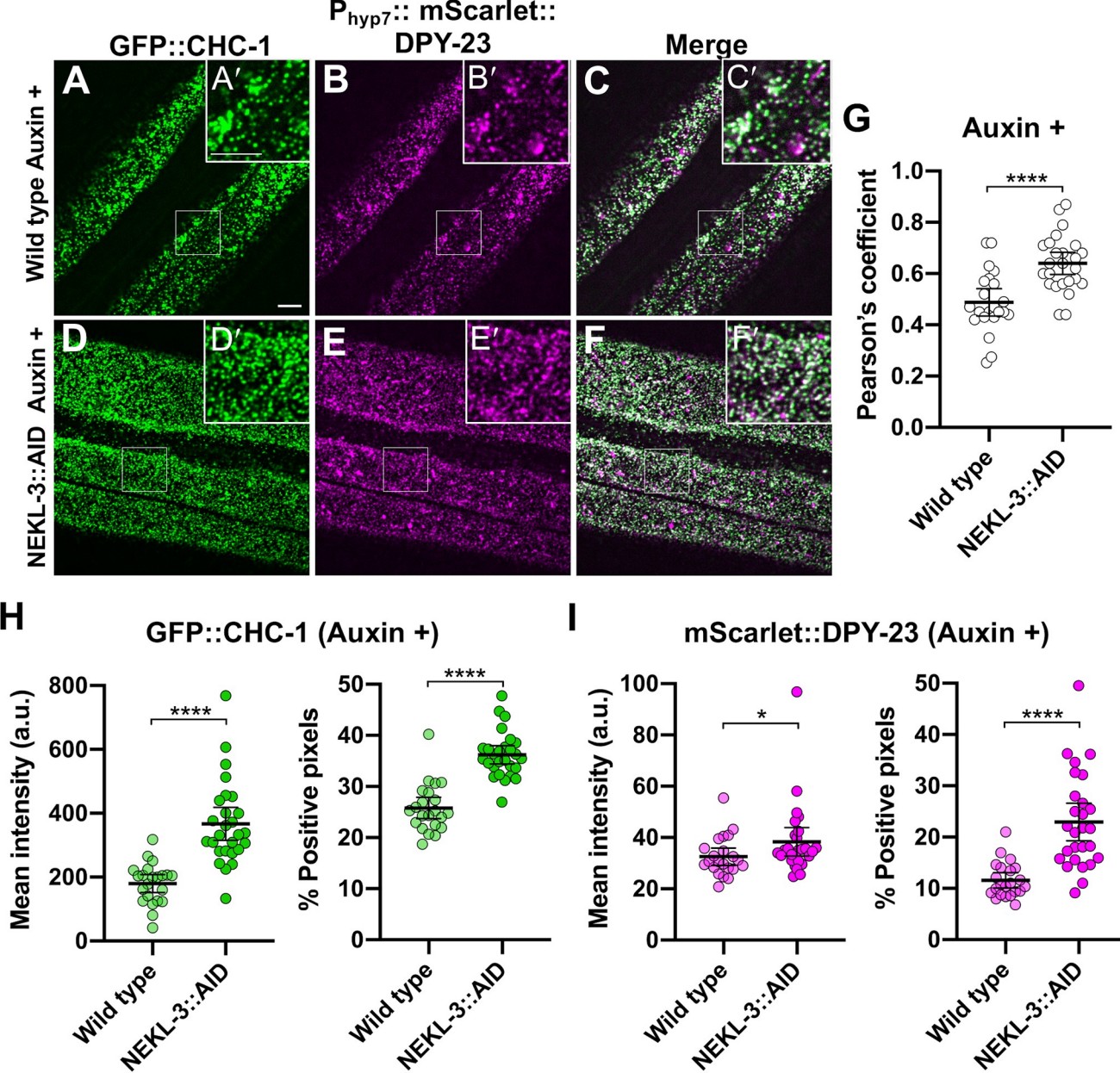

**Fig 7. AP2-associated clathrin is mislocalized in NEKL-depleted adults.** (A–F) Representative images of (A,D) GFP::CHC-1, (B,E) P$_{hyp7}$mScarlett::DPY-23/μ2, and merged images (C,F) in auxin-treated (A–C) wild-type and (D–F) NEKL-3::AID adults. mScarlet is represented as magenta, overlap is white. (A'–F') Magnified inserts of indicated regions in (A–F). Bar sizes in A and A' = 5 μm (for A–F and A'–F'). (G) Pearson's r coefficients are shown for the indicated strains; circles correspond to images from individual worms. (H,I) Mean intensities and the percentage of positive pixels above threshold were calculated for (H) GFP::CHC-1 and (I) mScarlet::DPY-23/μ2 for auxin-treated wild-type and NEKL-3::AID adults. p-Values were determined using two-tailed Mann-Whitney tests; ****$p < 0.0001$, *$p < 0.05$. Raw data are available in S1 File.

treated animals, a 1.5-fold reduction relative to auxin-treated wild type (Fig 8C, 8F and 8G, S4 Fig, S10 Movie). Thus, loss of either NEKL-2 or NEKL-3 results in a strong reduction in clathrin exchange. We note that some bleached spots exhibited homogenous recovery within the bleached area whereas others exhibited nonhomogeneous recovery. This was true for both NEKL-2::AID and NEKl-3::AID auxin-treated worms, although nonhomogeneous recovery was observed more frequently in NEKL-2::AID (16/21) than in NEKL-3::AID (5/17) samples.

Similar to what we observed for GFP::CHC-1 mean intensities and pixels above threshold (Fig 6), the AID tag appeared to cause a partial loss of NEKL-2 and NEKL-3 activities, even in the absence of auxin; the mobile fraction in untreated NEKL-2::AID and NEKL-3::AID worms was 61% and 49%, respectively (Fig 8B, 8D and 8F–8H, S4 Fig, S11 and 12 Movies). In addition, the more robust effects observed for NEKL-3::AID versus NEKL-2::AID adults are consistent with the stronger effects observed for NEKL-3::AID in the GFP::CHC-1 localization assays (Fig 6). Overall, our FRAP data demonstrate that NEKLs strongly affect clathrin exchange in the apical epidermis. Coupled together, our findings indicate that loss of NEKLs leads to the increased stability of clathrin structures located at or near the apical hyp7 surface, a defect consistent with reduced rates of clathrin uncoating and/or the formation of clathrin microcages.

## Clathrin defects in adults are suppressed by reduced activity of AP2

We next determined if clathrin defects associated with adult NEKL::AID loss in adults could be rescued by *fcho-1* and AP2 suppressor mutations. Consistent with our findings in *nekl-2; nekl-3* larvae (Fig 5), mutations in *fcho-1* and *apa-2/α* suppressed apical clathrin localization defects in auxin-treated NEKL-2::AID adults (Fig 9A and 9D). In fact, there were no statistically supported differences between auxin-treated and untreated adults with respect to GFP::CHC-1 mean intensities or the percentage of pixels above threshold in worms containing the *fcho-1* or *apa-2/α* suppressors (Fig 9B, 9C, 9E and 9F). Likewise, loss of *fcho-1* activity was found to strongly reduce effects on clathrin induced by auxin treatment of NEKL-3::AID strains, although the extent of suppression was less than that observed in NEKL-2::AID strains (S5 Fig).

In addition, we assayed GFP::CHC-1 mobility using FRAP in NEKL-2::AID strains containing the *fcho-1(fd296)* suppressor mutation. Untreated day-2 adults exhibited an average mobility of 72%, similar to what we observed for untreated wild type (74%; Figs 8A, 8F, 9G and 9H, S4 Fig). Most notably, auxin treatment had little or no effect on clathrin mobility in NEKL-2:: AID adults that contained *fcho-1(fd296)*; we observed a 1.1-fold reduction in mobility that was not statistically significant (Fig 9G and 9H). This is in strong contrast to the 1.5-fold reduction in mobility observed in auxin-treated NEKL-2::AID adults (p < 0.01; Figs 8G and 9H, S4 Fig). Moreover, one-way ANOVA did not indicate statistically significant differences between the GFP::CHC-1 mobilities of wild-type (Auxin +/−), NEKL-2::AID (Auxin−), and NEKL-2::AID *fcho-1(fd296)* (Auxin +/−) adults (p = 0.20). Likewise, loss of *fcho-1* activity strongly suppressed clathrin mobility defects in auxin-treated NEKL-3::AID adults (S5 Fig). Taken together, our results indicate that reduced AP2 activity strongly suppresses *nekl*-associated clathrin defects in both larvae and adults. Moreover, the suppression of clathrin defects in adults demonstrates that the restoration of normal trafficking in the suppressor strains is not merely a consequence of alleviating molting defects.

## Loss of NEKLs leads to defects in the trafficking of a physiologically relevant membrane cargo

The observed effects on clathrin localization and dynamics following NEKL::AID depletion led us to investigate the role of NEKLs in regulating plasma membrane cargo. LRP-1 is orthologous to human LDL receptor related protein 2 (LRP2), is expressed in hyp7, and is required for normal molting [15]. LRP-1 is thought to bind to and transport low-density lipoproteins into the epidermis from the extracellular space between the epidermis and cuticle. Once internalized, the release and breakdown of LDLs may provide a key source of sterols, which are required for generating hormonal cues necessary for the molting process [4, 13].

Consistent with previous reports, LRP-1::GFP was expressed in wild-type in a punctate pattern in the apical epidermis of larvae and adults (Fig 10A and 10C) [15, 64]. Moreover, these

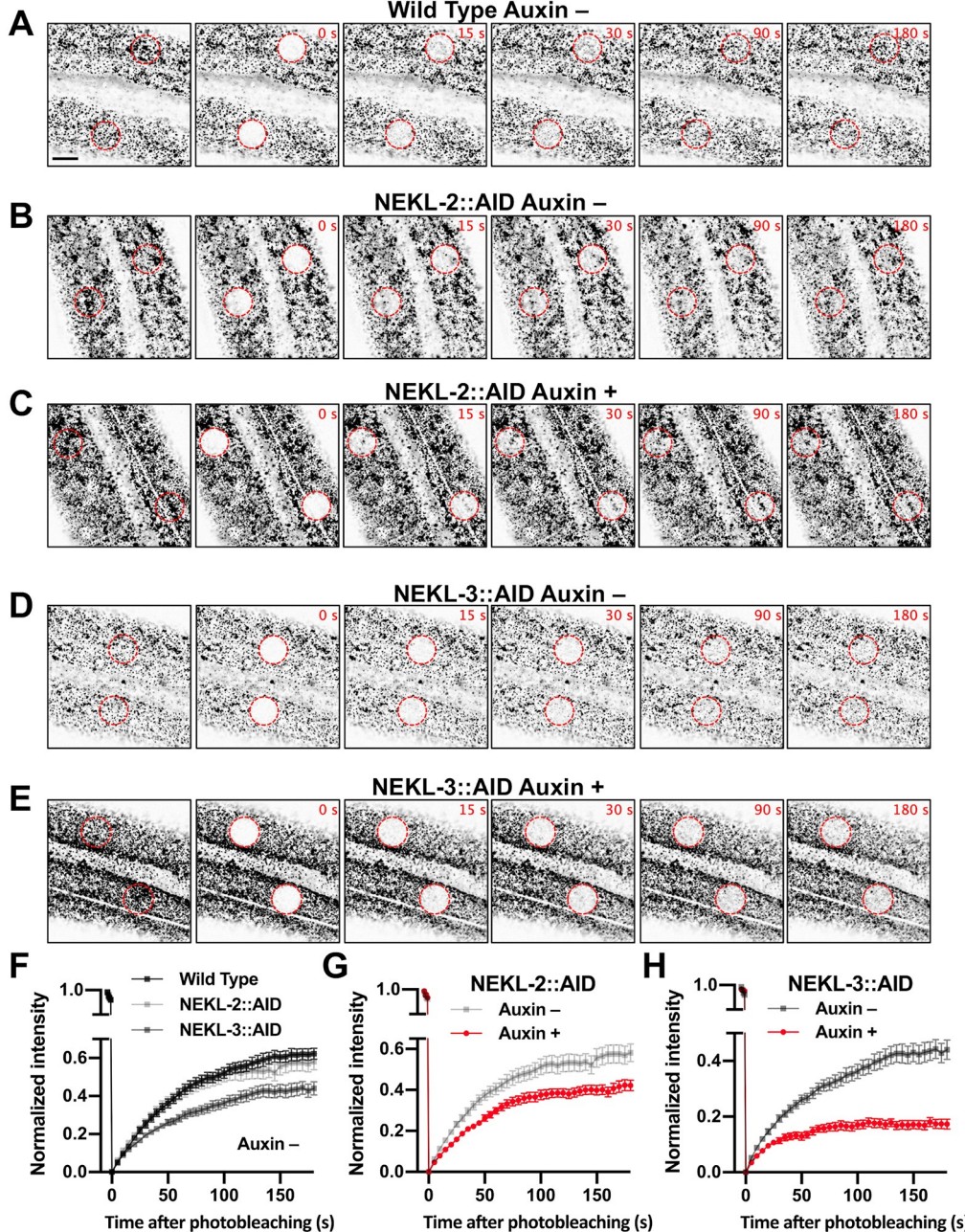

**Fig 8. Clathrin dynamics are altered in NEKL-depleted adults.** (A–E) Representative time-lapse confocal images of FRAP assays performed using the indicated strains in the presence and absence of auxin (20 h). Images show day-2 adults expressing GFP::CHC-1 within the apical region of the hyp7 epidermal syncytium. Background subtraction was performed using the same parameters for all images; minimum and maximum pixel values were kept consistent for all images. Inverted fluorescence was used to aid clarity. Each panel series contains images at the pre-bleach stage together with images at 0 s, 15 s, 30 s, 90 s, and 180 s after bleaching. Red dashed circles indicate the region of photobleaching. Bar size in A = 5 μm (for A–E). (F–H) Corresponding fluorescence recovery curves of GFP::CHC-1 after photobleaching. Normalized average mean intensities of the photobleached regions were plotted as a function of time using 5-s intervals; error bars denote SEM. (F) Fluorescence recovery curves for wild-type, NEKL-2::AID, and NEKL-3::AID adults in the absence of auxin. (G,H) Fluorescence recovery curves for NEKL-2::AID (G) and NEKL-3::AID (H) strains in the presence and absence of auxin. Raw data are available in S1 File.

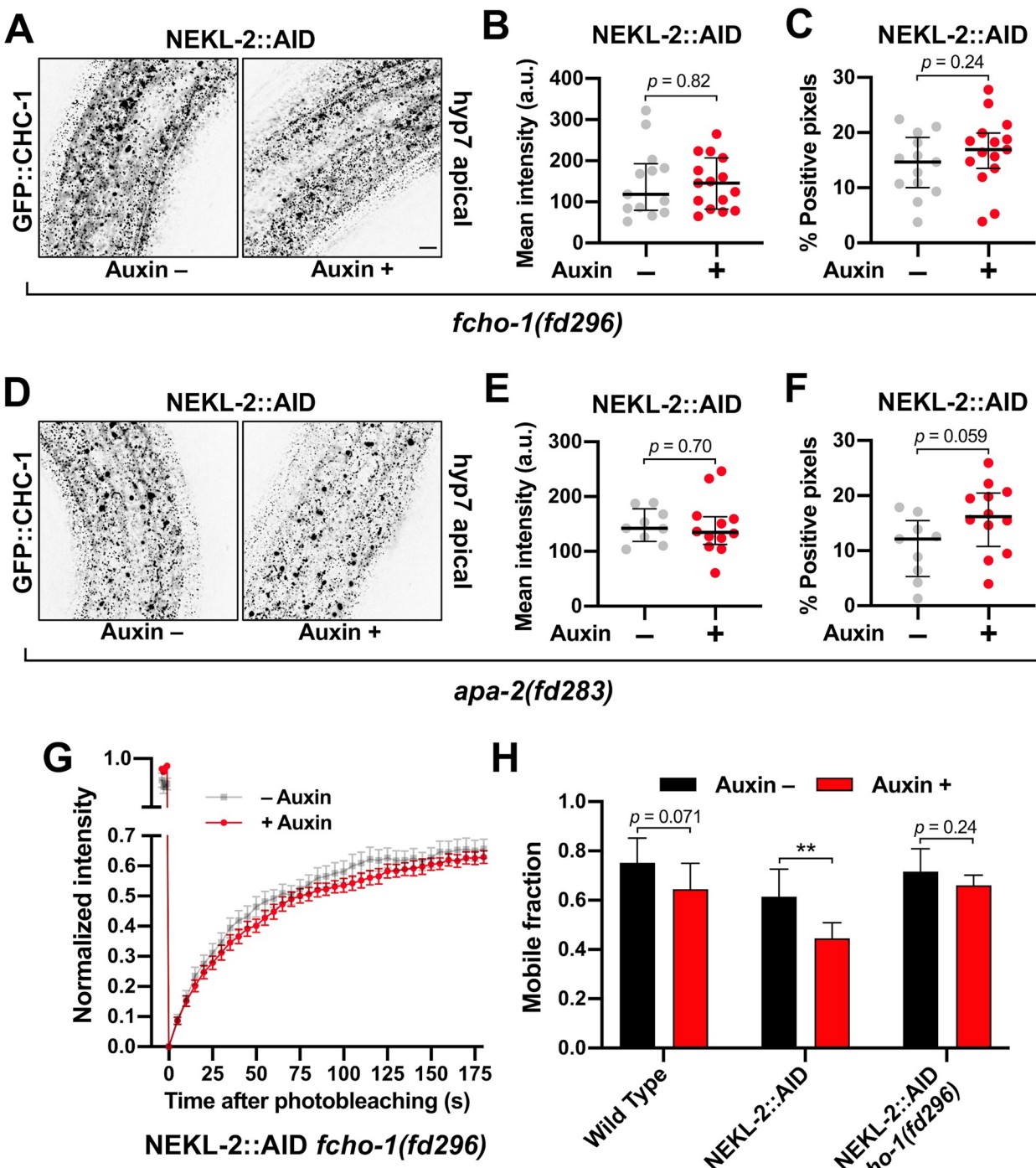

**Fig 9. NEKL-associated clathrin defects are rescued by loss of AP2 activity.** (A,D) Representative confocal images of day-2 adults expressing GFP::CHC-1 within the apical region of the hyp7 epidermal syncytium. Assays were performed on NEKL-2::AID animals containing null alleles of (A–C) *fcho-1(fd296)* and (D–F) *apa-2(fd283)* in the presence and absence of auxin (20 h). Background subtraction was performed using the same parameters for all images; minimum and maximum pixel values were kept consistent for all images. Inverted fluorescence was used to aid clarity. Bar size in A = 5 µm (for A and D). (B,C,E,F) For individual adults, the mean GFP::CHC-1 intensities (B,E) and the percentage of GFP-positive pixels above threshold (C,F) were determined. (G) Fluorescence recovery curves of NEKL-2::AID *fcho-1(fd296)* day-2 adults in the presence and absence of auxin. Normalized average mean intensities of photobleached regions were plotted as a function of time using 5-s intervals; error bars denote SEM. (H) Bar plot showing the mobile fractions from FRAP analyses of wild-type, NEKL-2::AID, and NEKL-2::AID *fcho-1(fd296)* adults. (B, C,E,F) The group mean and 95% confidence intervals (error bars) are shown. (H) Error bars indicate 95% confidence intervals. p-Values were determined using two-tailed Mann-Whitney tests; $**p < 0.01$]. Raw data are available in S1 File.

puncta correspond to clathrin-coated pits or vesicles based on co-localization with a marker for AP2 (S6 Fig). Notably, depletion of NEKL-3::AID with auxin led to a dramatic change in the localization pattern of LRP-1::GFP in the apical hyp7 region of adults (Fig 10A and 10B; S13 and S14 Movies). Specifically, LRP-1::GFP localization became highly diffusive in the plane of the apical membrane after NEKL-3::AID depletion. This effect was reflected by a 3-fold increase in the percentage of GFP-positive pixels above threshold in auxin-treated wild-type versus NEKL-3::AID adults (Fig 10A and 10B). Furthermore, no overlap in values between auxin-treated wild type and NEKL-3::AID worms was observed (Fig 10B), indicating that apical LRP-1::GFP localization provides a particularly strong readout for NEKL defects. Consistent with previous data, a modest increase in the percentage of positive pixels was observed in untreated NEKL-3::AID adults as compared with wild type (Fig 10B). In addition, we observed a modest increase in the percentage of LRP-1::GFP-positive pixels in medial planes of hyp7 in both treated and untreated NEK-3::AID adults relative to wild type (S6 Fig; S13 and S14 Movies). Together, our data suggest that NEKL-3 is required for the apical

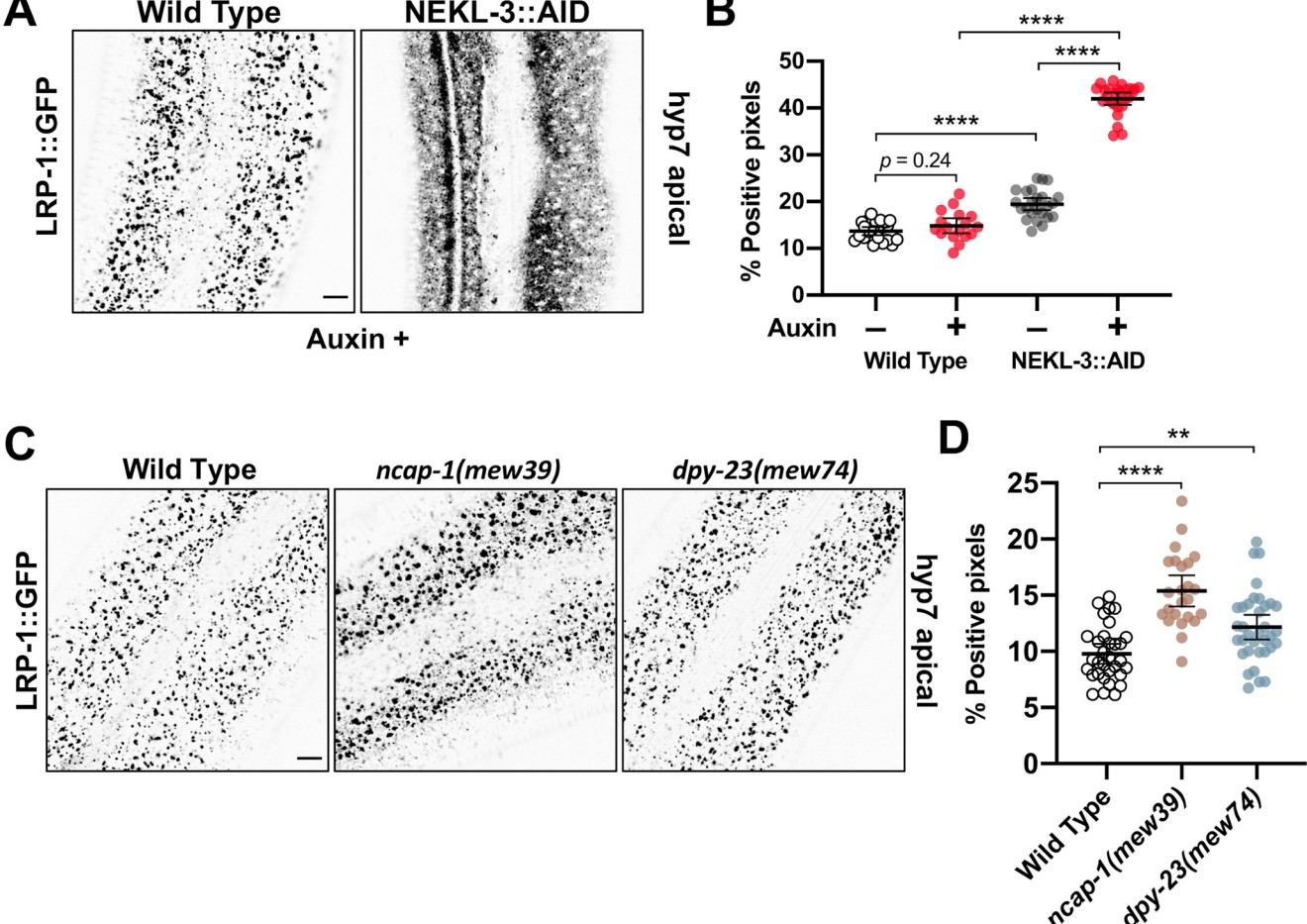

**Fig 10. Cargo trafficking is disrupted in NEKL-depleted adults.** (A,C) Representative confocal images of day-2 adults expressing LRP-1::GFP in the apical region of the hyp7 epidermal syncytium. Background subtraction was performed using the same parameters for all images; minimum and maximum pixel values were kept consistent for all images. Inverted fluorescence was used to aid clarity. Bar sizes in A,C = 5 μm. (B,D) The percentage of GFP-positive pixels above threshold was determined for individual auxin-treated (20 h) wild type and NEKL-3::AID adults (B) and untreated wild-type, *ncap-1(mew39)*, and *dpy-23(mew74)* (C) adults. The group mean and 95% confidence interval (error bars) are shown. p-Values were determined using two-tailed Mann-Whitney tests; ****$p < 0.0001$, **$p < 0.01$. Raw data are available in S1 File.

endocytosis of LRP-1–consistent with our findings of severe dysfunction in clathrin-coated pit/vesicle dynamics upon loss of NEKL-3.

## NEKLs affect trafficking through a mechanism that is distinct from that of NCAP-1

Several pieces of genetic data suggest that the NEKLs could function through a molecular mechanism that is similar to that of NCAP-1. For example, mutations in *ncap-1* and the *nekls* all suppress the phenotype of AP2 loss-of-function mutants (Fig 4) [42], as do mutations in *dpy-23*/μ that promote the open AP2 conformation [40]. Furthermore, loss of *ncap*-1 or *dpy-23*/μ open mutants strongly enhanced molting defects in partial loss-of-function *nekl* backgrounds (Fig 3), which can occur when proteins act in a common pathway or process.

To determine if NCAP-1 and the NEKLs might act through a similar mechanism, we first examined CHC-1::GFP in *ncap-1(mew39)* null mutants. Although mean levels of apical GFP:: CHC-1 were increased by 1.2-fold in *ncap-1(mew39)* mutants (Fig 11A and 11B), these changes were far less pronounced than the ~2- or 3-fold increases observed between auxin-treated wild-type and NEKL::AID adults (Fig 6). Moreover, no statistical difference was detected in the percentage of GFP-positive pixels between wild-type and *ncap-1(mew39)* worms (Fig 11C). Consistent with a lack of strong effects observed for *ncap-1* mutants, we detected no differences in GFP::CHC-1 mean intensity and pixels above threshold between wild type and *dpy-23(mew74)* open mutants (Fig 11A–11C). Likewise, we failed to detect statistical differences in clathrin localization in *ncap-1(mew39)*; *dpy-23(mew74)* double mutants, although there appeared to be a trend towards a reduced number of the larger clathrin puncta and an increased number of smaller puncta in this strain (Fig 11A–11C).

As an additional test, we examined GFP::CHC-1 recovery after photobleaching in *ncap-1 (mew39)* and *dpy-23(mew74)* mutants. Notably, whereas NEKL::AID auxin-treated worms showed a pronounced reduction in the mobile GFP fraction relative to wild type (Fig 8, S4 Fig), we observed no significant differences between wild-type, *ncap-1(mew39)*, and *dpy-23(mew74)* worms (Fig 10D–10F). Thus, an increase in the level of open/active AP2 does not appear to result in a marked change in the mobility of clathrin at the apical plasma membrane in hyp7.

We next examined LRP-1::GFP localization in *ncap-1(mew39)* and *dpy-23(mew74)* adults. Although we observed a modest increase in the percentage of pixels above threshold (1.2- to 1.5-fold) in these strains relative to wild type, the effects were again much weaker than what we observed in auxin-treated NEKL-3::AID worms (Figs 10C, 10D and 6). These findings strongly suggest that the NEKLs act through a mechanism that is fundamentally different than NCAP-1 and are therefore unlikely to regulate the conformation of AP2 (e.g., to promote the closed state). Nevertheless, our collective data demonstrate that the NEKLs affect a trafficking process that is highly sensitive to the balance between open and closed AP2 conformations.

## The mammalian NEKL-3 orthologs, NEK6 and NEK7, rescue molting and clathrin defects associated with *nekl-3* loss

The human NIMA kinase (NEK) family of proteins includes two closely related orthologs of NEKL-3, NEK6 and NEK7, which are ~70% identical and ~85% similar to NEKL-3 and ~80% identical and 90% similar to each other [20]. Given their high degree of conservation, we wanted to determine if the human homologs could rescue molting and clathrin-associated defects in *nekl-3* loss-of-function backgrounds.

We generated transgenes expressing human *NEK6* and *NEK7* cDNAs under the control of the wild-type *nekl-3* promoter. Our constructs included C-terminal fusions of the *NEKs* to an intron-containing GFP cassette as well as cDNA-only constructs that lack the GFP tag. Notably,

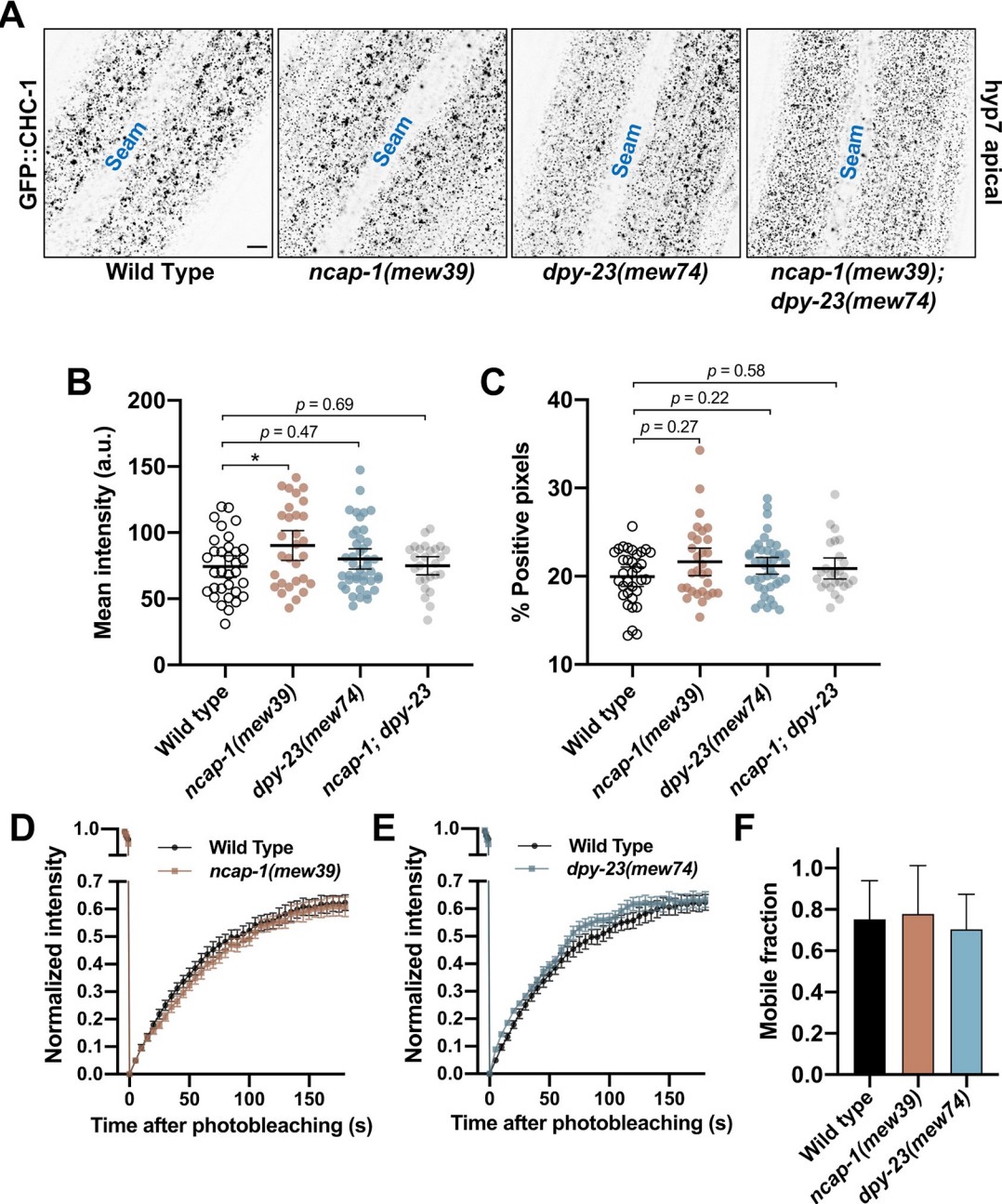

**Fig 11. NEKLs and NCAP-1 likely affect trafficking through distinct mechanisms.** (A) Representative confocal images of day-2 adults expressing GFP::CHC-1 within the apical region of the hyp7 epidermal syncytium. Background subtraction was performed using the same parameters for all images; minimum and maximum pixel values were kept consistent for all images. Inverted fluorescence was used to aid clarity. Bar size in A = 5 µm. (B,C) For individual adults, the mean GFP::CHC-1 intensity (B) and the percentage of GFP-positive pixels above threshold (C) were determined. The group mean and 95% confidence interval (error bars) are shown. p-Values were determined using two-tailed Mann-Whitney tests; *p < 0.05. (D,E) Fluorescence recovery curves of GFP::CHC-1 after photobleaching showing *ncap-1(mew39)* (D) and *dpy-23(mew74)* (E) day-2 adults. Normalized average mean intensities of photobleached regions were plotted as a function of time using -5 s intervals; error bars denote SEM. (F) Bar plot showing the percent mobile fraction from FRAP analyses; error bars indicate 95% confidence intervals. Raw data are available in S1 File.

both $P_{nekl-3}$::NEK6::GFP and $P_{nekl-3}$::NEK7::GFP were able to rescue molting defects in null *nekl-3(gk506)* and hypomorphic *nekl-3(sv3)* mutants, although their level of rescue was weaker than what was obtained using a ~5-kb region derived from the *nekl-3* genomic locus (Fig 12A; S7 Fig) [20]. In addition to full rescue to the adult stage, which was observed in ~50% of *nekl-3 (gk506)* worms expressing human NEK::GFP transgenes (Fig 12A), many transgene-positive worms exhibited at least partial rescue based on their size and stage of arrest (S7 Fig).

A *NEK6* cDNA-only construct ($P_{nekl-3}$::NEK6) also provided significant rescue in both backgrounds, albeit to a lesser degree than the *NEK6::GFP* fusion (Fig 12A; S7 Fig). In contrast, a *NEK7* cDNA-only construct ($P_{nekl-3}$::NEK7) failed to rescue molting defects in either background (Fig 12A; S7 Fig). The reduced rescuing activity of both the *NEK6* and *NEK7* cDNA-only constructs is likely due to weak expression, as intron-containing markers boost the expression of cDNAs in *C. elegans* and other systems [65, 66]. Consistent with our rescue data for *nekl-3* mutants, $P_{nekl-3}$::NEK6, but not $P_{nekl-3}$::NEK7, was able to partially rescue molting defects following auxin treatment of NEKL-3::AID strains, although rescue was again less robust than for wild-type *nekl-3* (Fig 12B).

We next tested if wild-type *nekl-3* and $P_{nekl-3}$::NEK6 could rescue GFP::CHC-1 localization defects in NEKL-3::AID strains. As shown in Fig 6, auxin-induced NEKL-3::AID depletion led to a 1.8-fold increase in mean apical GFP::CHC-1 intensity relative to untreated NEKL-3::AID worms. Furthermore, as expected, expression of the wild-type NEKL-3 protein fully rescued clathrin defects in auxin-treated NEKL-3::AID worms (Fig 12C, 12E and 12F). Specifically, no increase was observed in the mean intensity of GFP::CHC-1 or in the percentage of pixels above threshold in treated versus untreated NEKL-3::AID—*nekl-3*⁺ worms. Notably, NEKL-3:: AID worms expressing NEK6 displayed only a 1.2-fold increase in the mean intensity of GFP:: CHC-1 relative to untreated NEKL-3::AID—*NEK6*⁺ controls (Fig 12D) and this fold change was significantly lower than NEKL-3::AID worms containing no transgene (Fig 12E). NEK6 expression did not, however, rescue the percentage of positive pixels above threshold relative to the no-transgene control (Fig 12D and 12F). We note that direct comparisons of GFP:: CHC-1 localization data in Figs 6 and 12 were not possible because of differences in marker composition and thresholding procedures. Nevertheless, our findings demonstrate that both human NEK6 and NEK7 can rescue molting defects in *nekl-3* mutants and that NEK6 can partially rescue clathrin defects in strains with reduced NEKL-3 function. These findings strongly suggest that the trafficking functions demonstrated for NEKL-3 are conserved across species.

## Discussion

### NEKLs regulate clathrin-mediated endocytosis

In this study, we have demonstrated the NEKLs to be novel regulators of clathrin-mediated endocytosis. More specifically, depletion of NEKL-2 or NEKL-3 at the adult stage led to a strong increase in the levels of clathrin and to a dramatic decrease in the mobility of clathrin at the apical surface of hyp7. These findings demonstrate that endocytic defects following NEKL depletion occur independently of molting defects and are thus not merely a secondary consequence of defective molting. The physiological relevance of our findings is further bolstered by our approach to studying NEKLs within their native context and within an intact developing organism. Moreover, given the requirement for endocytic trafficking factors in the molting process [4], our findings indicate that molting defects in *nekl–mlt* mutants are likely to be a consequence of abnormal trafficking.

Additional evidence to support a direct role of the NEKLs in trafficking includes our finding that both molting and trafficking defects associated with reduced NEKL–MLT activity are strongly suppressed by mutations that decrease the levels of open/active AP2. This includes

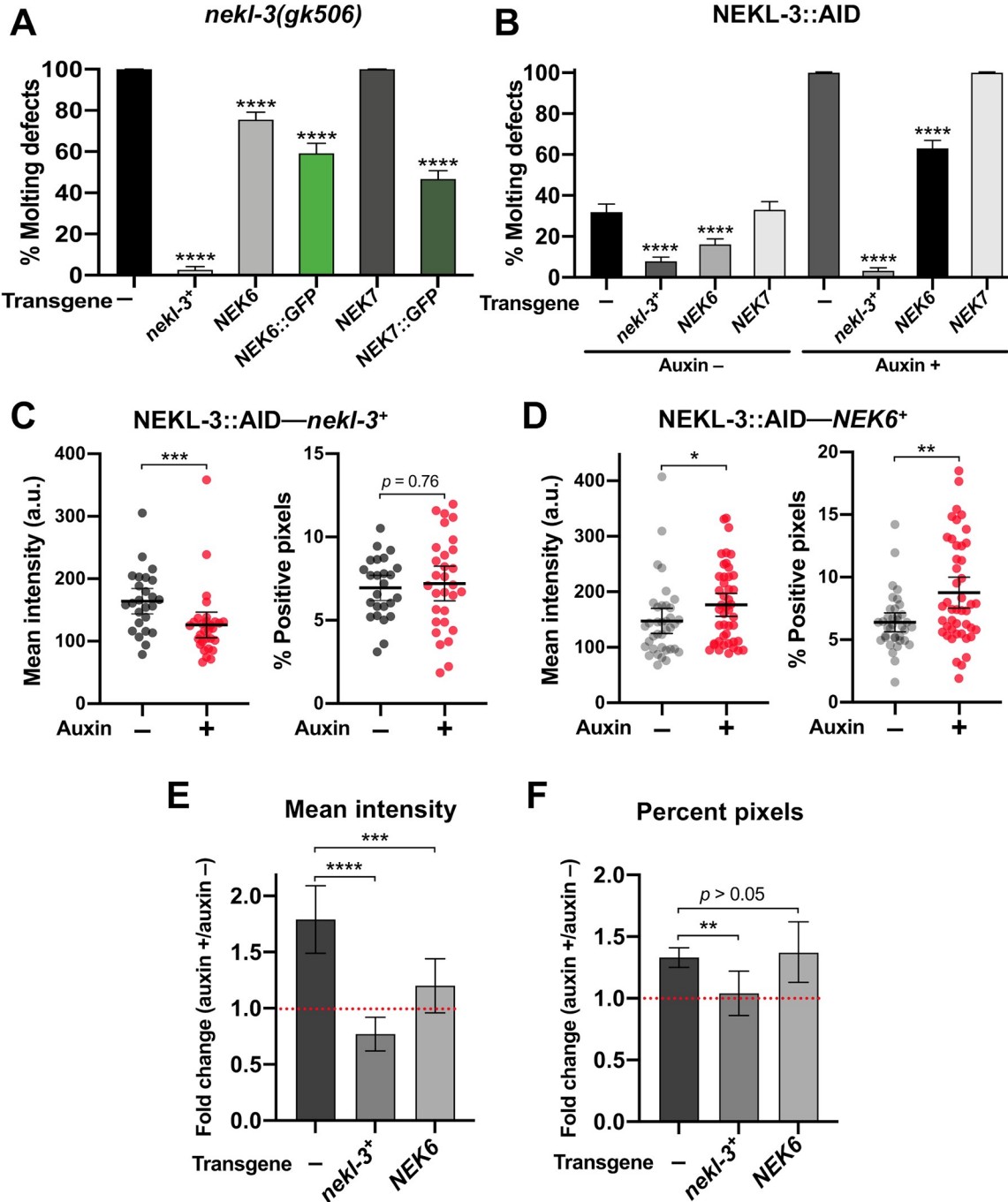

**Fig 12. The human orthologs of *nekl-3*, NEK6 and NEK7, rescue molting and trafficking defects.** (A,B) Bar plots showing rescue of molting defects in *nekl-3(gk506)* (A) and NEKL-3::AID (B) strains with the indicated transgenes. *NEK6::GFP* and *NEK7::GFP* refer to $P_{nekl-3}$::*NEK6*::*GFP* and $P_{nekl-3}$::*NEK7*::*GFP*, respectively. *NEK6* and *NEK7* refer to $P_{nekl-3}$::*NEK6* and $P_{nekl-3}$::*NEK7*, respectively. p-Values were determined using Fischer's exact test; $^{****}p < 0.0001$. (C,D) For individual adults, the mean GFP::CHC-1 intensity (left graphs) and the percentage of GFP-positive pixels above threshold (right graphs) were determined for animals expressing wild-type *nekl-3* (C) or *NEK6* (D). The group mean and 95% confidence interval (error bars) are shown. p-Values were determined using two-tailed Mann-Whitney tests; $^{*}p < 0.05$, $^{**}p < 0.01$, $^{***}p < 0.001$. (E,F) Fold-changes (ratios) of auxin-treated (20 h) versus untreated day-2 NEKL-3::AID adults expressing the indicated transgenes were determined for apical hyp7 GFP::CHC-1 mean intensities (E) and the percentage of GFP-positive pixels above threshold (F). Error bars indicate 95% confidence intervals. The dashed red line at 1.0 indicates no change in auxin-treated versus untreated worms. Statistical analyses for ratios were carried out as described in the Materials and Methods; $^{**}p < 0.01$, $^{***}p < 0.001$, $^{****}p < 0.0001$. Raw data are available in S1 File.

loss-of-function mutations of individual AP2 subunits as well as mutations in an allosteric activator of AP2, FCHO-1. Likewise, mutations that increase the levels of open/active AP2 strongly enhanced *nekl–mlt* defects, including *dpy-23*/μ open mutants and loss of function of NCAP-1, an allosteric inhibitor of AP2. Additionally, defects in clathrin localization and mobility after NEKL-2 and NEKL-3 depletion in adults were rescued by mutations that decrease the levels of open/active AP2. Finally, depletion of NEKL-3 in adults led to dramatic defects in LRP-1/LRP2 endocytosis, which is required for normal molting and is internalized in multiple systems via clathrin-mediated endocytosis [64, 67]. Taken together, our findings indicate that loss of NEKLs leads to defects in the internalization of cargo that is required for normal molting.

Given our specific findings for AP2 and FCHO-1, a question arises as to whether mutations in other genes connected to early steps of endocytosis can suppress defects in *nekl–mlts*. Notably, loss of function in *itsn-1*/intersectin and *ehs-1*/EPS15, which have been shown to physically and functionally interact with FCHO–AP2 [29, 48, 53, 68–70], failed to suppress molting defects in *nekl-2; nekl-3* mutants (S8 Fig). In fact, mutations in *itsn-1* and *ehs-1* were found to enhance the penetrance of molting defects following partial loss of *nekl–mlt* function by RNAi (S8 Fig). Likewise, similar enhancement effects were observed in strains containing mutations in other core clathrin-pathway components including *unc-11*/PICALM, *unc-57*/endophilin A2, *sel-5*/AAK1, and *lst-4*/SNX9/18/33 (S8 Fig). These results indicate that *nekl–mlts* are suppressed specifically by a reduction in AP2 activity and not by the generic impairment of the endocytic pathway. In addition, these findings are consistent with our model that depletion of NEKL activity reduces the uptake of plasma membrane cargo (e.g., LRP-1), a defect that can be further aggravated by inhibiting other components of the endocytic pathway.

## Determining the role of NEKLs in clathrin-mediated endocytosis

One explanation to account for a number of our genetic and cell biological observations is that the NEKLs could promote the closed state of AP2. However, several observations argue against this model. (1) In contrast to clathrin localization in *nekl* mutants and NEKL-depleted adults, clathrin localization in *ncap-1* and *dpy-23*-open mutants was largely unaffected. (2) Unlike clathrin mobility in NEKL-depleted adults, clathrin mobility in *ncap-1* and *dpy-23*-open mutants was indistinguishable from that of wild type. (3) Defects in LRP-1 localization in *ncap-1* and *dpy-23*-open mutants were much less severe than those observed after depletion of NEKL-3. (4) We failed to observe molting defects in *ncap-1(mew39)*, *dpy-23(mew74)* and *ncap-1 (mew39)*; *dpy-23(mew74)* double mutants, indicating that increased levels of open/active AP2 alone are not sufficient to induce molting defects. These observations indicate that the NEKLs act through a mechanism that is distinct from that of NCAP-1. Although it remains possible that the NEKLs may carry out multiple functions in endocytosis, including a role in AP2 regulation, we favor a model whereby the NEKLs control a trafficking step that is highly sensitive to the balance between open and closed AP2 conformations.

Based on our FRAP and localization data of clathrin, we suggest that the NEKLs may promote the uncoating of clathrin from internalized coated vesicles. Disassembly of the clathrin coat following membrane scission is carried out by the conserved uncoating ATPase Hsc70, together with its co-chaperone(s), auxilin/GAK [29, 71]. Auxilin bound to clathrin recruits Hsc70 to clathrin-coated vesicles, and Hsc70 then interacts with the clathrin heavy chain to alter the conformation of the triskelia, leading to coat disassembly. Notably, inhibition of DNJ-25, the *C. elegans* ortholog of auxilin, leads to increased clathrin accumulation within the hyp7 epidermis, decreased clathrin mobility (in coelomocytes), and molting defects [62], all of which are observed in *nekl–mlt* mutants. These observations are consistent with the NEKLs

acting either in parallel or upstream of Hsc70–auxilin to promote uncoating, models that will be tested in future studies.

Importantly, we observed that altering the balance between open and closed AP2 conformations led to strong genetic modulation of the molting and clathrin-associated phenotypes in *nekl–mlt* mutants. How might altering the balance of open/closed AP2 impact the efficiency of clathrin uncoating? It is well established that the conformational opening of AP2 promotes clathrin assembly at membranes and the subsequent formation of coated pits [26, 27, 29, 30]. As such, one appealing hypothesis is that the conformational closing of AP2 may in turn facilitate the uncoating of clathrin from internalized vesicles. Although this kind of reciprocal function for AP2 in both the coating and uncoating of clathrin has not, to our knowledge, been directly demonstrated, it is consistent with much of our genetic and cell biological data.

Notably, auxilin can bind to both clathrin and AP2 [72], providing a possible mechanism by which uncoating could be coupled to the conformation of AP2. In addition, it has been suggested that clathrin uncoating may require the disruption of contacts between AP complexes and clathrin [73] and that clathrin and AP2 uncoating are linked [74]. Furthermore, the levels of PIP2 in membranes affect clathrin uncoating [75–77], despite the absence of direct binding of clathrin to membranes. In contrast, AP2 contains a PIP2-binding site that is available for membrane interactions only in the open conformation [27, 38, 39], suggesting that the conformation of AP2 could affect clathrin uncoating.

A number of early studies using cell culture systems, however, demonstrated that clathrin and AP2 uncoating/exchange can take place largely independently of each other. For example, clathrin release can occur in the absence of AP2 release from membranes [78–81]. In addition, cytosolic AP2 can be exchanged with membrane-bound AP2 in vesicles containing a stabilized clathrin coat [74, 82, 83]. Moreover, we failed to observe reduced recovery of clathrin after photobleaching in *ncap-1* and *dpy-23*-open mutants, suggesting that, in an otherwise wild-type background, shifting the AP2 balance toward the open/active conformation does not detectably alter the kinetics of clathrin exchange. Additional studies will be necessary to elucidate the precise mechanism behind AP2 suppression of *nekl–mlt* phenotypes.

We also note that that the reduced mobility of clathrin observed in NEKL-depleted strains could be due to the formation of structures termed "clathrin microcages". Microcages are extremely small, sharply curved, unusual polymers of clathrin that lack any membrane [84]. Microcages have been observed in mammalian cells that overexpress an inactive form of Hsc70 [85], when potassium or ATP has been depleted from cells, or when cells are exposed to hypertonic sucrose [63, 84]. Similar to what we observed in NEKL-depleted worms, the fluorescence recovery of clathrin after photobleaching was strongly impaired in cells with abundant microcages [63, 74]. These microcages may in fact sequester cytosolic clathrin, leading to reduced clathrin availability and lower levels of endocytosis. Consistent with this, conditions that induce microcages also lead to the dispersal of LDL receptors on the membrane surface [84], a phenotype we observed for the LDL receptor, LRP-1, in NEKL-depleted strains. Future EM studies will determine the ultrastructure of the exchange-resistant clathrin foci in NEKL-depleted worms.

## Trafficking functions of NIMA kinase family members are likely conserved across species

Our rescue of *nekl-3* molting defects with constructs expressing human NEK6 and NEK7 strongly indicates that these proteins carry our similar functions in diverse organisms. Moreover, we observed rescue of clathrin mislocalization defects in *nekl-3* mutants with human NEK6. Although the large majority of papers on NEK6 and NEK7 have focused on functions

associated with cell division, a high-throughput analysis of the human kinome by Zerial and colleagues indicated that clathrin-mediated endocytosis is strongly decreased in cells when *NEK6* or *NEK7* is targeted by siRNAs [86]. Likewise this study also identified moderate defects in endocytosis following siRNA treatment of *NEK8* and *NEK9*, the closest human homologs of *nekl-2*. A second large-scale study by the Zerial group also reported abnormal trafficking after knockdown of NEK6 and NEK8 based on multiple parameters [87]. In addition, there is some evidence to suggest that NIMA family members in *A. nidulans* and *S. cerevisiae* carry out functions connected to endocytosis [88]. Lastly, protein association studies of NEK6 and NEK7 have identified factors connected to both trafficking (e.g., AP2A1 and AP2B1) and to cytoskeletal factors that impact endocytic functions (e.g., CDC42 and actin) [89–91], although the functional significance of these interactions was not explored.

Given these collective observations, we propose that the control of intracellular trafficking may be an ancient and conserved function for members of the NIMA kinase family. This would represent a role for mammalian NEK kinases that has been largely overlooked but could be relevant to their involvement in diseases including cancer and ciliopathies [23, 92–106]. Future studies in both *C. elegans* and mammalian systems will be important to establish the precise functions of NIMA-related kinases in cellular trafficking and human disease processes.

## Materials and methods

### Strains and maintenance

*C. elegans* strains were maintained according to standard protocols [107] and were propagated at 22˚C. Strains used in this study are listed in S2 Table.

### CRISPR/Cas9

CRISPR/Cas9 ribonucleoproteins in combination with the *dpy-10* co-CRISPR method [108–110] was used to generate genomic lesions except for the *pw27(nekl-2::aid)* and *pw29(nekl-3::aid)* alleles, which were created using the self-excising cassette method [111]. *pw27(nekl-2::aid)* and *pw29(nekl-3::aid)* guide RNA and repair template plasmids were created from existing plasmids pDD268 and pDD268, respectively [21], using Gibson assembly to directly replace mNeonGreen sequences with AID sequences. Cas9 enzyme was purchased from University of California, Berkeley, whereas, crRNA, tracrRNA, and the repair templates were brought from GE Healthcare Dharmacon, Inc. Briefly, 7.8 µl Berkeley Cas9 (6.4 µg/µl), 0.75 µl 3 M KCl, 0.75 µl 200 mM HEPES (pH 7.4), 5 µl 0.17 mM tracrRNA, 0.8 µl 0.3 mM *dpy-10* crRNA, 2 µl 0.3 mM target crRNA, and 0.75 µl of distilled water were mixed and incubated at 3˚C for 15 minutes. After incubation, 16 µl of *dpy-10* repair template and 1.6 µl of 10 µM target repair template was added, and the mixture was injected into the gonads of adult worms of the desired strains. See S1 Text for information on the sequences of oligos used for all CRISPR/Cas9 studies; sequence information on the obtained CRISPR/Cas9 lesions is indicated in S1 Table.

### Western blot analysis

L4 animals were picked to NGM plates with or without 1mM auxin and harvested after 20 hrs at 20˚C. For each lane 100 young adult animals of each genotype/condition were handpicked into 30 µl of lysis buffer (100 mM Tris pH 6.8, 8% SDS, 20 mM β-mercaptoethanol) and incubated at 37˚C for 60 min in a thermocycler. Then 5 µl of 6X Laemmli sample buffer was added, and the samples were further incubated at 100˚C for 5 min. Extracted proteins were separated by 4–20% gradient ExpressPlus PAGE gel (Genscript) and blotted to nitrocellulose using

Genscript MOPS buffer. After blocking in 5% non-fat milk for 60 min, the blot was rinsed and probed with mouse Monoclonal ANTI-FLAG M2 (Sigma, 1:1000 dilution) followed by HRP-conjugated Goat anti-mouse antibody (1:10,000). Signals were detected using Supersignal West Pico chemiluminescent substrate (Thermo Scientific) and exposure to film. The blot was then stripped and probed with rabbit anti-actin antibody (Sigma A2066, 1:1000) followed by HRP-conjugated Goat anti-rabbit antibody (1:10,000).

### NEK transgenic rescue

Transgenic strains used in these studies were obtained by microinjecting ~100 ng/μl of the plasmid of interest and ~50 ng/μl of pTG96.2 (*sur-5*::*RFP*) into worm gonads [112]. Plasmids for NEK-6::GFP and NEK-7::GFP expression were generated by inserting a ~2 kb promoter region from *nekl-3* (LGX 12391550–12393472) into pPD95.75 using PstI and XbaI restriction sites (pDF225). cDNAs for NEK6 and NEK7 were inserted into pDF225 using XbaI and KpnI sites to generate pDF241 and pDF244, respectively. The GFP cassette was removed from pDF241 and pDF244 by digesting with KpnI and EcoRI, blunting with T4 DNA Polymerase, and re-ligating to give plasmids pDF421 (NEK6) and pDF422 (NEK7).

### Marker transgenes

Single copy miniMos transgenes were generated to express markers for plasma membrane clathrin-coated pits (DPY-23/APM-2/μ2) and Trans-Golgi Network clathrin-coated pits (APM-1/μ1) in the hypodermis [113]. Genomic DNA for *apm-1*, and a published minigene for *dpy-23*, was cloned into a customized version of miniMos vector pCFJ910, including the hyp7-specific promoter from gene Y37A1B.5 ($P_{hyp7}$), red fluorescent protein wrmScarlet-I (mScarlet), and the 3'UTR from gene *let-858* [40, 114, 115].

### RNAi

dsRNAs corresponding to *apa-2*, *dpy-23*, *aps-2*, *ncap-1*, and *fcho-1* were generated using standard methods [116], followed by injection at 0.8–1.0 μg/μl into worm gonads (see S1 Text for oligo sequences). For molting enhancement studies, RNAi feeding was performed using bacterial strains from Geneservice following standard protocols [117]. Worm strains were first grown for one (Fig 3B and 3D) or two (Fig 3E) generations on *lin-35(RNAi)* plates to increase RNAi susceptibility [118]. Gravid adults were then transferred to experimental RNAi plates and allowed to lay eggs for ~24 h, and F1 progeny were scored for molting defects after an additional ~72 h. Because *mlt-3(RNAi)* induced a high percentage of molting defects in wild type, *mlt-3(RNAi)* bacteria were diluted 1:10 with the control *gfp(RNAi)* strain.

### Image acquisition

Fluorescence images were acquired using an Olympus IX81 inverted microscope with a Yokogawa spinning-disc confocal head (CSU-X1). Excitation wavelengths were controlled using an acousto-optical tunable filter (ILE4; Spectral Applied Research). MetaMorph 7.7 software (MetaMorph Inc.) was used for image acquisition. z-Stack images were acquired using a 100×, 1.40 N.A. oil objective; FRAP time-lapse images were acquired using a 60×, 1.35 N.A. oil objective. DIC images were acquired using a Nikon Eclipse epifluorescence microscope using 10×, 0.25 N.A. and 40×, 0.75 N.A. objectives. Image acquisition was controlled by Openlab 5.0.2 software (Agilent Inc.). Animals were immobilized using 0.1 M levamisole in M9 buffer. Supplementary movies 1–6 and 13, 14 are Z-stack images (0.2 μm steps).

For FRAP assays, anesthetized worms were analyzed immediately after being placed on slides (<10 min). The apical region of hyp7 was brought into focus, and a circular spot (7.272 μm in diameter) was photobleached using an iLas2 system (BioVision Technologies) with a 56-ms pulse of a 405-nm laser set at 50% power. After photobleaching, GFP::CHC-1 FRAP was detected by imaging every 5 s for a total of 180 s. Data acquired for Figs 8 and 11 were carried out at the same time using the same microscope and acquisition procedures. For this reason, some identical control data is present in these figures. Supplementary movies 7–12 are 200 s long and have been condensed 25× (8 s).

## Image analysis

To quantify the mean intensity and the percentage of fluorescence-positive pixels above threshold (Figs 5,6 and 8–11), background fluorescence was subtracted from z-stack images using Fiji software (NIH) available at https://imagej.net/Fiji/Downloads). For a given z-plane of interest, the polygon selection tool was used to demarcate the region of hyp7 followed by mean intensity measurement. The percentage of fluorescence-positive pixels for the region of interest was determined after thresholding, and the same thresholding algorithm was used for strain comparisons.

FRAP time-lapse images were aligned to correct for any movement of the worms by using the "Rigid Body" Transformation method in the StackReg plugin (available at https://imagej.net/StackReg). Next, the time-lapse images were background subtracted and analyzed using the custom-written Stowers plugin (available at http://research.stowers.org/imagejplugins), which can be used with the Fiji software. The photobleached region was selected and mean intensities were quantified before photobleaching and after photobleaching for each time point. Fitting of FRAP curves was performed using batch FRAP fit in the Stowers plugin. Fitted curves were then normalized to prebleach mean intensities and then averaged to obtain final recovery curves. Mobile fractions were quantified using the values obtained from the fitted curves. The mobile fraction is determined by the following equation: amplitude/(prefrap-baseline) (see https://research.stowers.org/imagejplugins/ImageJ_tutorial2.html).

## Auxin treatment experiments

Auxin was purchased as indole-3-acetic acid from Alfa Aesar. In these experiments, L4-stage worms were transferred to plates and left to develop into adults (~20 h). A 0.4 M (100×) stock auxin solution was made by dissolving 0.7 g of auxin in 10 mL 100% ethanol. Each plate containing day-1 adults was treated with a mixture of 25 μl of stock auxin solution and 225 μl of distilled water.

## Statistics

Statistical tests were performed as indicated using software from Prism GraphPad. Statistical tests comparing fold change ratios (Fig 12; S5 Fig) were carried out as described by Fay and Gerow [119].

## Supporting information

**S1 Fig. Clathrin localization in NEKL::AID-depleted larvae.** Representative images of untreated (Auxin−) and auxin-treated (Auxin +; 20 h) NEKL-2::AID and NEKL-3::AID arrested larvae expressing GFP::CHC-1. Inverted fluorescence images are shown to aid clarity. Background subtraction was performed using the same parameters for all images; minimum

and maximum pixel values were kept consistent for all images. Bar in upper left panel = 5 μm (for all panels).
(TIFF)

**S2 Fig. Supplemental GFP::CHC–1 expression data.** (A–F) Representative images of untreated (Auxin–) and auxin-treated (Auxin +; 20 h) wild-type day-2 adults expressing GFP:: CHC-1. (G–L) Representative images of similarly treated NEKL-2::AID (G–I) and NEKL-3:: AID (J–L) day-2 adults expressing GFP::CHC-1. Inverted fluorescence was used to aid clarity. Background subtraction was performed using the same parameters for all images; minimum and maximum pixel values were kept consistent for all images. Bar in A = 5 μm (for all panels). Mean GFP::CHC-1 intensities (B,E,H,K) and the percentage of GFP-positive pixels above threshold (C,F,I,L) were determined for day-2 adults. Panels A–C show data for the apical region of hyp7; panels D–L show data for a medial region of hyp7. (B,C,E,F,H,I,K,L) Both the group mean and 95% confidence interval (error bars) are shown. Note that wild-type showed small but statically significant increases in medial GFP::CHC-1 intensity and pixels above threshold after exposure to auxin, suggesting that auxin itself could exert a weak effect on GFP::CHC-1 localization. p-Values were determined using two-tailed Mann-Whitney tests; $^{**}p < 0.01$, $^{*}p < 0.05$. Raw data are available in S1 File.
(TIFF)

**S3 Fig. AP1-associated clathrin is reduced in NEKL-3-depleted adults.** (A–F) Representative images of (A,D) GFP::CHC-1, (B,E) $P_{Y37A1B.5}$APM-1::mScarlet, and merged images (C,F) in auxin-treated (A–C) wild-type and (D–F) NEKL-3::AID adults. Bar size in A = 5 μm (for A– F). (G) Pearson's r coefficients are shown for the indicated strains; circles correspond to images from individual worms. (H,I) p-Values were determined using student's T-Test; $^{*}p < 0.05$. Raw data are available in S1 File.
(TIFF)

**S4 Fig. Supplemental NEKL::AID FRAP data.** (A,B) Fluorescence recovery curves for wild-type (A,B), NEKL-2::AID (A), and NEKL-3::AID (B) day-2 adults in the presence and absence of auxin (20 h). Analyses were carried in the apical hyp7 region with GFP::CHC-1. Normalized average mean intensities of the photobleached regions were plotted as a function of time using 5-s intervals; error bars denote SEM. (C) Mobile fractions from FRAP data in panels A and B; error bars show 95% confidence intervals. (D) p-Values for all possible comparisons for data in panel C were determined using two-tailed Mann-Whitney tests. Raw data are available in S1 File.
(TIFF)

**S5 Fig. Loss of FCHO-1 activity partially suppresses NEKL-3::AID defects.** Mean GFP:: CHC-1 intensities (A) and the percentage of GFP-positive pixels above threshold (B) were determined for individual adults. (C) Comparative fold changes for mean intensities (M.I.) and positive pixels above threshold (P.P) are shown for the indicated genotypes in the presence (+) and absence (–) of auxin. (D) Fluorescence recovery curves of NEKL-3::AID *fcho-1(fd296)* day-2 adults in the presence and absence of auxin. Normalized average mean intensities of photobleached regions were plotted as a function of time using 5-s intervals; error bars denote SEM. (E) Bar plot showing the mobile fractions from FRAP analyses of wild-type, NEKL-3:: AID, and NEKL-3::AID *fcho-1(fd296)* adults. (C,F) Error bars show 95% confidence intervals. The dashed red line at 1.0 indicates no change in auxin-treated versus untreated worms. Statistical analyses for ratios (C) were carried out as described in the Materials and Methods; $^{**}p < 0.01$, $^{****}p < 0.0001$. Raw data are available in S1 File.
(TIFF)

**S6 Fig. Supplemental LRP-1 data.** (A) Representative images showing strong colocalization of LRP-1::GFP and a marker for AP2, P$_{dpy-7}$::mScarlet::DPY-23. mScarlet is represented as magenta, overlap is white. (B) Representative confocal images of LRP-1::GFP in auxin treated wild-type and NEKL-3::AID adults. Images show the medial plane of hyp7 (also see S13 and S14 Movies). Inverted fluorescence was used to aid clarity. Bar size in A,B = 5 μm. (C) Percentage of GFP-positive pixels above threshold were determined for individual adults. p-Values were determined using two-tailed Mann-Whitney tests; ****$p < 0.0001$. Raw data are available in S1 File.
(TIFF)

**S7 Fig. Supplemental NEK rescue data.** (A,B) Bar plot showing rescue of molting defects in *nekl-3(sv3)* strains with the indicated transgenes. NEK6::GFP and NEK7::GFP refer to P$_{nekl-3}$::NEK6::GFP and P$_{nekl-3}$::NEK7::GFP, respectively. NEK6 and NEK7 refer to P$_{nekl-3}$::NEK6 and P$_{nekl-3}$::NEK7, respectively. p-Values were determined using Fischer's exact test; ****$p < 0.0001$. (B) Bar plot showing the percentage of L1/L2 versus L2–L4 arrested larvae in transgene-positive *nekl-3(gk506)* mutants. Note that ~40–60% of transgene-positive arrested larvae bypass the L1/L2 arrest point, whereas the large majority of transgene-minus worms arrest at L1/L2. Given ~50% rescue to adulthood by the NEK6::GFP and NEK7::GFP transgenes (Fig 12A), partial-to-full rescue occurs at a frequency of ~75% in transgene-positive *nekl-3(gk506)* mutants.
(TIFF)

**S8 Fig. Inhibition of many early-endocytic pathway genes enhance *nekl* defects.** (A) Bar plot showing failure to suppress molting defects in *nekl-2(fd81); nekl-3(gk894345)* double mutants by RNAi of *ehs-1*, *itsn-1*, and *ehs-1; itsn-1*, using dsRNA injection methods. (B–E) RNAi feeding of the *nekl-2*, *nekl-3*, and *mlt-2* was carried out in the indicated backgrounds. Error bars indicate 95% confidence intervals; p-values were determined using Fischer's exact test where proportions were compared to the wild-type allele. ****$p < 0.0001$, Raw data are available in S1 File.
(TIFF)

**S1 File. This excel file contains the raw data used for all quantitative data panels presented in Figs 1–11, including supplementary Figs.**
(XLSX)

**S1 Text. This MS Word file contains information describing the generation of all CRISPR alleles used in this study including sgRNAs, repair templates, and sequencing oligos.**
(DOCX)

**S2 Text. This MS Word file contains detailed information regarding specific author contributions for each figure.**
(DOCX)

**S1 Table. This MS Word file contains relevant genomic sequencing data of all CRISPR alleles generated in this study.**
(DOCX)

**S2 Table. List of all *C. elegans* strains used in this study.**
(DOCX)

**S1 Movie. Z-stack of wild-type (RT3402) day-2 adults; GFP::CHC-1; untreated.**
(AVI)

**S2 Movie. Z-stack of wild-type (RT3402) day-2 adults, GFP::CHC-1; auxin treated (20 h).**
(AVI)

**S3 Movie. Z-stack of NEKL-2::AID (RT3607) day-2 adults, GFP::CHC-1; auxin treated (20 h).**
(AVI)

**S4 Movie. Z-stack of NEKL-3::AID (RT3608) day-2 adults, GFP::CHC-1; auxin treated (20 h).**
(AVI)

**S5 Movie. Z-stack of NEKL-2::AID (RT3607) day-2 adults; GFP::CHC-1; untreated.**
(AVI)

**S6 Movie. Z-stack of NEKL-3::AID (RT3608) day-2 adults, GFP::CHC-1; untreated.**
(AVI)

**S7 Movie. FRAP of wild-type (RT3402) day-2 adults, GFP::CHC-1; untreated.**
(AVI)

**S8 Movie. FRAP of wild-type (RT3402) day-2 adults, GFP::CHC-1; auxin treated (20 h).**
(AVI)

**S9 Movie. FRAP of NEKL-3::AID (RT3608) day-2 adults, GFP::CHC-1; auxin treated (20 h).**
(AVI)

**S10 Movie. FRAP of NEKL-2::AID (RT3607) day-2 adults, GFP::CHC-1; auxin treated (20 h).**
(AVI)

**S11 Movie. FRAP of NEKL-2::AID (RT3607) day-2 adults, GFP::CHC-1; untreated.**
(AVI)

**S12 Movie. FRAP of NEKL-3::AID (RT3608) day-2 adults, GFP::CHC-1; untreated.**
(AVI)

**S13 Movie. Z-stack of wild-type (LH191) day-2 adults, LRP-1::GFP; auxin treated (20 h).**
(AVI)

**S14 Movie. Z-stack of NEKL-3::AID (WY1562) day-2 adults, LRP-1::GFP; auxin treated (20 h).**
(AVI)

## Acknowledgments

We thank Amy Fluet for editing and the G. Hollopeter Laboratory for strains and scientific input. Some strains were provided by the CGC, which is funded by NIH Office of Research Infrastructure Programs (P40 OD010440).

## Author Contributions

**Conceptualization:** Braveen B. Joseph, Vladimir Lažetić, Barth D. Grant, David S. Fay.

**Data curation:** Braveen B. Joseph, Yu Wang, David S. Fay.

**Formal analysis:** Braveen B. Joseph, Yu Wang, Phil Edeen, Vladimir Lažetić, Barth D. Grant, David S. Fay.

**Funding acquisition:** David S. Fay.

**Investigation:** Braveen B. Joseph, Yu Wang, Phil Edeen, Vladimir Lažetić, Barth D. Grant, David S. Fay.

**Methodology:** Braveen B. Joseph, Yu Wang, Phil Edeen, Vladimir Lažetić, Barth D. Grant, David S. Fay.

**Project administration:** Barth D. Grant, David S. Fay.

**Resources:** David S. Fay.

**Supervision:** Barth D. Grant, David S. Fay.

**Validation:** Braveen B. Joseph, Yu Wang, David S. Fay.

**Visualization:** Braveen B. Joseph, Yu Wang, David S. Fay.

**Writing – original draft:** Braveen B. Joseph, Barth D. Grant, David S. Fay.

**Writing – review & editing:** Braveen B. Joseph, Vladimir Lažetić, Barth D. Grant, David S. Fay.

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
