## [Decision Letter · Decision Letter 0]

7 Oct 2019

Dear Dr Fay,

Thank you very much for submitting your Research Article entitled 'Control of clathrin-mediated endocytosis by NIMA family kinases' to PLOS Genetics. Your manuscript was fully evaluated at the editorial level and by independent peer reviewers. The reviewers appreciated the attention to an important problem, but raised some substantial concerns about the current manuscript. Based on the reviews, we will not be able to accept this version of the manuscript, but we would be willing to review again a much-revised version. We cannot, of course, promise publication at that time.

Should you decide to revise the manuscript for further consideration here, your revisions should address the specific points made by each reviewer. In particular, Reviewer 1 raises an important concern about specificity of the suppression phenotypes, and the Reviewers shared concerns about the sub-cellular resolution and interpretation of the clathrin imaging experiments.  An editorial comment also raised by several of the reviewers: the text and references would benefit from editing to shorten the manuscript and increase its accessibility.  We will require a detailed list of your responses to the review comments and a description of the changes you have made in the manuscript.

If you decide to revise the manuscript for further consideration at PLOS Genetics, please aim to resubmit within the next 60 days, unless it will take extra time to address the concerns of the reviewers, in which case we would appreciate an expected resubmission date by email to plosgenetics@plos.org.

[LINK]

We are sorry that we cannot be more positive about your manuscript at this stage. Please do not hesitate to contact us if you have any concerns or questions.

Yours sincerely,

Jeremy Nance

Associate Editor

PLOS Genetics

Gregory P. Copenhaver

Editor-in-Chief

PLOS Genetics

Reviewer's Responses to Questions

**Comments to the Authors:**

Reviewer #1: Joseph et al. report on a potential endocytic role for the NIMA-like kinases NEKL-2 and NEKL-3. The general function of these kinases has not been well-described, although this group has published several papers reporting the molting defects and suppression of the NEKL phenotypes by CDC42 and related actin regulators. The manuscript largely relies on a genetic analysis of the NEKL phenotype and suppression of the phenotype by reducing the function of AP2 subunits or the AP2 activator FCHO. The manuscript is a genetic tour-de-force: the number of alleles, CRISPR-driven domain mutations, and careful measurement of phenotypic penetrances and backgrounds is impressive. Certainly the degree to which the manuscript uses genetics makes it an excellent candidate for PLoS Genetics. There is very little cell biology and no biochemistry, which makes me somewhat uncomfortable with the specificity of some of the manuscript’s conclusions. It is an absolutely huge manuscript in every respect (10 figures, 8000 word main text, 139 references) – I know PLoS Genetics has no word limit, but some editing for conciseness would make the manuscript much more approachable. Statistics and N numbers appear to be well-done and reported. In the end, I am concerned about some of the reported interpretations, so would like to know more about how specific the AP2 suppression of the NEKL mutants is, as well as additional details of the Clathrin:GFP and LRP:GFP localization.

1) Perhaps the biggest concern is that the authors strongly tie NEKL function to CME/AP2 due to the functional suppression of the NEKL phenotype. As the authors state, various perturbations of membrane trafficking pathways (for example, even perturbation of ER-Golgi trafficking) can cause molting defects. If NEKL or other molting mutants simply reflect a sensitivity of molting to trafficking defects, I would be concerned about how directly the authors tie NEKL function to clathrin uncoating (a very specific model based largely, but not only, on genetics). To test this, does general disruption of endocytosis suppress NEKL phenotypes, or does decreasing exocytosis (exocyst mutants, secretory Rab mutants) enhance these phenotypes? Would Dynamin mutants produce a similar AP2-like suppression? The NCAP-1 data is interesting, but since it acts in opposition to FCHO-1, these data do not appear to address this issue.

2) Along with the above, the second main part of the argument for NEKL controlling clathrin uncoating are the alterations in clathrin levels and LRP localization. I am sure that there are limitations in the worm system, but the degree of characterization of these defects is somewhat poor. The authors should point out what Clathrin structures they think are aberrantly changing. It appears that there are quite large, intense puncta, which I would be doubtful are true surface endocytic CCPs due to their size. There are other, much less intense, and hard-to-make-out signals that may constitute the major increase in signal in NEKL mutants. Is this true? What are the structural changes? Are the larger (or smaller) puncta Golgi or endosomes? How deep of a focal plane are the authors sampling? A dextran-uptake experiment would be a good addition and would reveal whether endocytic rates are changed, and if the dextran is trapped in the “still caged” clathrin vesicles. Also, does sec-23 cause similar changes in Clathrin localization? Absent some refinement of the data as suggested in points #1 and #2 I would be concerned about how molecularly-specific the authors' model of NEKL function is.

3) Similarly to the above, please point out the relevant LRP:GFP puncta – the manuscript says these puncta are “likely clathrin-coated pits” or “endosomes”, but this isn’t shown. Can colocalization with Clathrin or AP2 or some CME marker not be done? Even dextran-uptake could partially address this. The degree to which there is inference of compartmental identity based on little-to-no data is concerning.

4) The NEKL-AID experiments use wild-type as a control – is there a generic AID-containing strain that could be used for a more appropriate control?

5) The Discussion, like the rest of the manuscript, is extensive. I am not sure that the connection between CDC42 and clathrin uncoating is appropriate. Although CDC42-driven F-actin certainly has been implicated in scission and early endocytic events, these do not necessarily connect to the uncoating phenotype (or the authors should feel free to suggest more specifics on this).

Minor notes:

a) As noted above, the manuscript is enormous. It would be improved if it was edited down a bit. The Introduction meandered, and it wasn’t always clear why certain topics were being discussed. It also didn’t seem necessary to include some of the discussed information.

b) Please consider including the relevant mammalian homologs next to the worm gene names throughout, or in many more places – I was constantly going back and needing to look up which endocytic protein a particular worm gene encoded for.

Reviewer #2: Control of clathrin-mediated endocytosis by NIMA family kinases

This work corresponds to a follow-up study of the role and positioning of the NIMA family kinases which the Fay lab had initially characterized. Here the authors position the NEKL-2/3 proteins within the endocytic pathway by showing (i) that nekl-2; nekl-3 lof molting defects can be suppressed by mutations affecting AP-2 subunits of proteins required for AP-2 activation; (ii) conversely that mutations in proteins favoring AP-2 closed conformation enhance the phenotype of a weal nekl-2 or nekl-3; (iii) that nekl-2: nekl-3 loss affects clathrin recycling and induces plasma an accumulation of lipoprotein LRP-1 in vesicles or the plasma membrane apically. The authors conclude that NEKL-2/3 act in the endocytic pathway to favor through a process that remains to be characterized at the molecular level AP-2 closed conformation, perhaps by favoring clathrin uncoating from internalized vesicles.

I enjoyed reading this very well documented and written manuscript. I believe that it should be of interest to a wide audience and I therefore enthusiastically recommend for publication once some issues have been solved.

The main issue that the authors should address is to perform one more experiment to test their prediction regarding the role of NEKLs in uncoating clathrin from internalized coated vesicles. This would require EM (which they mention) to define is uncoated vesicles accumulate, or probably easier to implement examine the genetic interactions between nekl-2/3 lof and auxilin and/or Hsc70.

Another recommendation would be to speculate on the potential molecular target of NEKLs and avenues to test them. If needed the discussion on the actin phenotype of nekl-2/3 mutants could be trimmed down since actin is not brought up in the results but was part of a previous study. Of note, some v-ATPase subunits also undergo a complete reorganization during molting.

Minor points:

- Please number pages

- Please make a separate supplementary table of all strains (it is currently very difficult to read in the M&M)

- Consider reducing the number of references (nice to have a complete listing, but it takes room – I leave it to the Editor to decide)

- An s missing at the very top of page 20 (in the PDF downloaded from the PLoS website) in “statically“

- In movie S4, the recovery after FRAP does not appear to be homogeneous since there are clear dots appearing in a background of slow recovery, which I didn’t spot in other conditions. Could this tell us something about NEKL-2 function?

Reviewer #3: The manuscript by Joseph et al describes the role of the NEKL kinases in regulating endocytosis in the C. elegans epidermis. They first use a lethal phenotype to demonstrate the genetic interactions between nekl-2/-3 and AP2 subunits or AP2 regulators (fcho-1 and ncap-1), using in particular precise gene editing by CRISPR/Cas9; the results are then confirmed with the analysis of the Jowls phenotype. These genetic interactions are further confirmed by examining the localisation of an endogenous reporter for clathrin and an auxin-based degron system. To better understand how NEKL kinases control clathrin dynamics and endocytosis they next use FRAP experiments and examine the localisation of LRP-1, an apical transmembrane protein. They conclude that NEKL kinases regulates an AP2 dependent trafficking step. Finally they show that human homologs can also be implicated in trafficking.

This a very well written manuscript, easy to follow, and most experiments are carefully done and analyzed. In particular the use of gene editing and endogenously expressed markers makes the results quite convincing. The conclusions of the study are well supported by the experimental data.

I only have a few points to raise:

1) The most important one is the quantification of clathrin and LRP-1 localisation. A punctate staining can be tricky to analyze based solely on mean intensity and % of positive pixels, as you can in theory get similar results with very different patterns. If some results are quite convincing (eg Fig5), others are less obvious (Fig9C and Fig10A). For instance in Fig 10 the authors found no difference in clathrin localisation while the pictures they show look different between control and the ncap-1;dpy-23 double mutant. I recommend that the authors use a more detailed quantification method, analyzing the number of puncta, their intensity and their size in all ambiguous situations as this could reveal more subtle changes.

2) While the expression pattern of NEKL-2 and NEKL-3 has been described by the same group previously (Lažetić and Fay, 2017), the authors do not attempt to colocalize these proteins with either clathrin or AP2 subunits. Given their conclusions on NEKL-2/-3 possible function, it would be interesting to observe a possible colocalization.

3) FRAP controls look very similar in Fig 7F and Fig 10D-E. If they are identical it should be mentioned, and the reason for using the same control justified in the M&M section.

Minor points:

Remove a ; in the CRISPR § of the M&M section.

Briefly describe the western blot conditions in M&M

As far as I am aware the original apb-1 loss of function phenotype was first described by Shim et al 2000 in an MBoC paper which could be added to the references.

**Have all data underlying the figures and results presented in the manuscript been provided?**

Reviewer #1: Yes

Reviewer #2: Yes

Reviewer #3: Yes

PLOS authors have the option to publish the peer review history of their article (what does this mean?). If published, this will include your full peer review and any attached files.

Reviewer #1: No

Reviewer #2: Yes: Michel Labouesse

Reviewer #3: Yes: Grégoire Michaux

---

## [Decision Letter · Decision Letter 1]

28 Jan 2020

Dear Dr Fay,

We are pleased to inform you that your manuscript entitled "Control of clathrin-mediated endocytosis by NIMA family kinases" has been editorially accepted for publication in PLOS Genetics. Congratulations!

Yours sincerely,

Jeremy Nance

Associate Editor

PLOS Genetics

Gregory P. Copenhaver

Editor-in-Chief

PLOS Genetics

Comments from the reviewers (if applicable):

Reviewer's Responses to Questions

**Comments to the Authors:**

Reviewer #1: The revised manuscript from Joseph et al. is nicely improved – in my opinion, the authors respond in depth and appropriately to the reviewer issues raised. The manuscript continues to be a genetic tour-de-force, with the number of alleles, CRISPR-driven domain mutations, and careful measurement of phenotypic penetrances and backgrounds being impressive. The authors are very thoughtful in their discussions, and I think the manuscript is appropriate for publication in PLOS Genetics. Thanks to the authors for the resubmission comments. My only small critique would be that I thought the added lines 411-414 didn’t add anything to the manuscript, and instead made me wonder what these different structures might be (with no explanation offered). They also obscured a nice summary sentence that was above, so I would consider removing them. Congrats to the authors on a very nice manuscript (and good luck performing EM – it will be very interesting to see the results!).

Reviewer #2: Although the authors could not properly address my concerns, at least they tried and I accept that sometimes experiments cannot be performed as hoped. On the other hand, they made great efforts to address those raised by R#1, and should be praised for this.

Altogether, the additional experiments strengthened their global conclusions that the NIMA-like kinases of C. elegans control clathrin-mediated endocytosis, independently of molting. I thus remain convinced that this work should be accepted for publication in PLoS Genetics as it addresses the role of these kinases in vivo in a physiological context.

Reviewer #3: This revised version of the Joseph et al manuscript is a significant improvement. I recognize that quantification of some markers can be difficult. I still think that there should be better ways to quantify punctate staining in C. elegans but this is probably beyond the scope of this manuscript and could be tackled by the community in the future. I also agree that given the result, there is no need to show the (absence of) colocalization between NEKL-2/3 and clathrin/AP-2.

Finally I agree with the other reviewers that the manuscript was too long and although it has already been shortened, I feel that it could be further edited to make it more easily readable.

**Have all data underlying the figures and results presented in the manuscript been provided?**

Reviewer #1: Yes

Reviewer #2: Yes

Reviewer #3: Yes

PLOS authors have the option to publish the peer review history of their article (what does this mean?). If published, this will include your full peer review and any attached files.

Reviewer #1: No

Reviewer #2: No

Reviewer #3: Yes: Grégoire Michaux

**Data Deposition**

http://datadryad.org/submit?journalID=pgenetics&manu=PGENETICS-D-19-01501R1

**Press Queries**

---

## [Editor Report · Acceptance letter]

12 Feb 2020

PGENETICS-D-19-01501R1 

Control of clathrin-mediated endocytosis by NIMA family kinases 

Dear Dr Fay, 

We are pleased to inform you that your manuscript entitled "Control of clathrin-mediated endocytosis by NIMA family kinases" has been formally accepted for publication in PLOS Genetics! Your manuscript is now with our production department and you will be notified of the publication date in due course.

With kind regards,

Matt Lyles

PLOS Genetics

On behalf of:
